# Simple Distillation for One-Step Diffusion Models

**Huaisheng Zhu**[1], **Teng Xiao**[3,4], **Shijie Zhou**[2],
**Zhimeng Guo**[1], **Hangfan Zhang**[1], **Siyuan Xu**[1], **Vasant Honavar**[1]
[1]Pennsylvania State University, [2] University at Buffalo
[3] Allen Institute for AI (AI2) [4] University of Washington
hvz5312@psu.edu

## Abstract

Diffusion models have established themselves as leading techniques for image generation. However, their reliance on an iterative denoising process results in slow sampling speeds, which limits their applicability to interactive and creative applications. An approach to overcoming this limitation involves distilling multistep diffusion models into efficient one-step generators. However, existing distillation methods typically suffer performance degradation or require complex iterative training procedures which increase their complexity and computational cost. In this paper, we propose Contrastive Energy Distillation (CED), a simple yet effective approach to distill multistep diffusion models into effective one-step generators. Our key innovation is the introduction of an unnormalized joint energy-based model (EBM) that represents the generator and an auxiliary score model. CED optimizes a Noise Contrastive Estimation (NCE) objective to efficiently transfers knowledge from a multistep teacher diffusion model without additional modules or iterative training complexity. We further show that CED implicitly optimizes the KL divergence between the distributions modeled by the multistep diffusion model and the one-step generator. We present results of experiments which show that CED achieves performance comparable to that of representative baselines for distilling multi-step diffusion models while maintaining excellent memory efficiency.

## 1 Introduction

Diffusion models have become the prominent method for image generation, capable of producing highly realistic and diverse outputs through a stable training process [18, 45, 49, 44]. Unlike Generative Adversarial Networks (GANs) [14] and Variational Autoencoders (VAEs) [26], diffusion models rely on an iterative denoising procedure that progressively refines an initial Gaussian noise into detailed images [18, 50]. This method, while effective, is computationally intensive, often requiring dozens or even hundreds of neural network evaluations. Consequently, the slow sampling speed significantly reduces the practicality of diffusion models in interactive, creative applications.

To address this issue, several efforts have aimed at reducing the sampling steps required in reverse diffusion processes [34, 36, 67, 68, 52]. These approaches still require multiple steps to generate images. To significantly improve the efficiency of diffusion models, one-step diffusion methods have been proposed. In particular, distillation techniques have become increasingly popular for one-step generation, achieving state-of-the-art performance [70]. One line of work is score-based distillation, which includes iteratively training an auxiliary score model with the one-step generator [35, 47, 61, 65] or using an additional discriminator to create a GAN [69, 64], both obtaining great results.

However, these methods heavily rely on auxiliary models and the iterative training of multiple components, leading to substantial GPU consumption, prolonged training times, and increased complexity due to the additional hyperparameters introduced by these extra components, which require careful tuning. In contrast, trajectory-based distillation offers a more lightweight alternative

39th Conference on Neural Information Processing Systems (NeurIPS 2025).

by progressively increasing the sampling intervals of the student model [1, 15, 31, 32, 46, 51, 29]. This strategy avoids the need for extra models and iterative training, significantly simplifying the training pipeline. Nevertheless, these techniques typically experience great performance degradation compared to the original diffusion models when generating images in just one step. Therefore, in this paper, it naturally raises a question: *How can we design a simple, efficient and effective distillation algorithm for one-step diffusion models without iterative training and additional components?*

In this paper, we approach the problem by introducing an unnormalized joint Energy-based Model (EBM) that models the auxiliary score model (commonly referred to as the "fake score model" in prior work) and the one-step generator. Our key innovation lies in utilizing this EBM to distill knowledge from a teacher diffusion model. The energy function is designed to directly quantify the discrepancy between images generated by the one-step generator and outputs from the fake score model. A central challenge of our framework is to enable efficient training without relying on additional modules or complex architectures. To address this issue, we propose Contrastive Energy Distillation (CED), a method that trains our

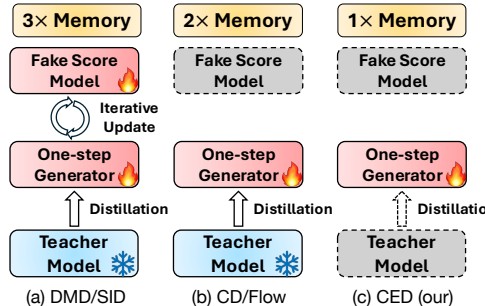

Figure 1: Training pipelines for different kinds of distillation methods. Our method (CED) uses only 1/3 or 1/2 the memory compared with them by loading fewer models during training.

joint energy-based model inspired by the Noise Contrastive Estimation (NCE) framework [16]. Specifically, we sample synthetic images from a pre-trained teacher diffusion model prior to training, and from the generator's previous training iteration to serve as negative samples. These are then used to optimize the NCE objective. This approach removes the need for extra components and iterative updates, enabling more resource-efficient training compared to previous methods, as shown in Figure 3. Furthermore, we show that, with a specific design of the energy function, our method implicitly optimizes the Kullback–Leibler (KL) divergence—an objective widely used in distribution matching distillation of one-step diffusion models. This provides a foundation for the effectiveness of our approach in transferring knowledge from multi-step diffusion models to a single-step generator.

In summary, the main contributions of this paper are: (i) We propose Contrastive Energy Distillation (CED), a simple and efficient distillation algorithm leveraging a contrastive loss on our joint energy-based model to transform multi-step diffusion models into a one-step generator. Inspired by noise contrastive estimation, CED eliminates the need for auxiliary modules, offering a novel and streamlined approach for designing one-step diffusion model distillation strategies. (ii) We demonstrate that CED implicitly optimizes the KL divergence for distribution matching distillation, providing a principled foundation for its distillation effectiveness. (iii) CED achieves competitive performance that also rely solely on one-step generators during training without auxiliary components.

## 2 Related Works

**Diffusion models** have become a powerful framework for generative modeling [18, 53, 50, 22], achieving remarkable success across a wide range of domains, including image generation [44, 42, 45], audio synthesis [27], video generation [10, 48], and molecular design [19, 62]. These models work by gradually transforming random noise into coherent data through a reverse diffusion process. Despite their state-of-the-art performance, the iterative nature of diffusion models leads to computational overhead, making them less suitable for real-world applications. To accelerate the sampling process of diffusion models, numerous techniques have been proposed to speed up generation through distillation. These approaches have achieved impressive results and can be broadly categorized into score-based distillation [35, 47, 61, 65, 69, 64] and trajectory-based distillation [52, 46, 32, 51, 36, 17, 63, 34, 43].

**Score-based distillation** methods were initially introduced in the context of text-to-3D synthesis [40, 56, 55], where a pre-trained text-to-image diffusion model is used to define a distribution matching loss. Inspired from these approaches, score distillation has been extended to the training of diffusion models themselves [65, 11, 35, 37]. These approaches either involve iteratively training an auxiliary score network alongside the one-step generator [35, 47, 61, 65], or incorporate an additional discriminator to form a GAN-like framework [69, 64] to further improve their performance. While effective, these methods depend heavily on auxiliary models and the iterative training of multiple

components, resulting in high GPU usage, longer training durations, and added complexity from extra hyperparameters associated with the additional components that require careful tuning.

**Trajectory-based Distillation** methods eliminate the need for additional modules and iterative training, which greatly simplify the overall pipeline. They offer a more streamlined approach by progressively increasing the sampling intervals of the student model [1, 15, 31, 32, 46, 51, 29]. Luhman et al. [34] precompute a dataset of image-noise pairs using an ODE sampler and train a student model to learn the mapping in a single step. Recified Flow simplifies the ODE trajectories to make them easier for one-step models to learn [31, 32]. Progressive Distillation eliminates the need for precomputed data by iteratively training student models, each reducing the number of sampling steps [46]. Consistency Distillation (CD) [51, 52] and TRACT [1] further improve training by enforcing self-consistency of the student's outputs along the ODE trajectory, aligning them with the teacher model. Moreover, Flow matching (FM) [31, 30] based methods distill themselves to shorten sampling steps. However, when used to generate images in a single step, these methods often suffer from significant performance degradation compared to the original diffusion models. Therefore, in this paper, we propose a simple, efficient, and effective approach to distilling a multistep diffusion model into a one-step generator using Contrastive Energy Distillation (CED). CED requires neither auxiliary models nor iterative training, and outperforms trajectory-based distillation methods, which rely on loading the teacher model during training. CED relies solely on the one-step generator during training, which make the process both simple and efficient.

## 3 Notations and Preliminaries

**Diffusion Models**. Given a dataset $\{\mathbf{x}^{(1)}, \ldots, \mathbf{x}^{(N)}\}$ sampled from a real data distribution $p_{\text{real}}(\mathbf{x})$, diffusion models learn to generate these samples by progressively denoising corrupted versions [18, 50]. In the forward diffusion process, noise is incrementally added to a sample $\mathbf{x} \sim p_{\text{real}}$ across $T$ timesteps until it becomes pure Gaussian noise. Formally, at each timestep $t$, the diffused samples follow $p_{\text{real},t}(\mathbf{x}_t) = \int p_{\text{real}}(\mathbf{x})q(\mathbf{x}_t \mid \mathbf{x})d\mathbf{x}$ with $q(\mathbf{x}_t \mid \mathbf{x}) \sim \mathcal{N}(\alpha_t \mathbf{x}, \sigma_t^2 \mathbf{I})$, where $\alpha_t, \sigma_t > 0$ are scalars determined by the noise schedule. During training, the model learns to reverse this corruption process by the distribution $q(\mathbf{x}_{t-1} \mid \mathbf{x}_t) \sim \mathcal{N}(\mathbf{x}_{t-1}; \mu(\mathbf{x}_t, t), \sigma_t^2 \mathbf{I})$, which predict a denoised estimate, $\mu(\mathbf{x}_t, t)$, conditioned on the timestep $t$ and the noisy sample $\mathbf{x}_t$. By iterative refinement, the model recovers images resembling those drawn from $p_{\text{real}}$. Once trained, the denoised estimate aligns with the gradient of the likelihood function (i.e., the score function) of the diffused distribution:

$$s_{\text{real}}(\mathbf{x}_t, t) = \nabla_{\mathbf{x}_t} \log p_{\text{real},t}(\mathbf{x}_t) = -\frac{\mathbf{x}_t - \alpha_t \mu_{\text{real}}(\mathbf{x}_t, t)}{\sigma_t^2}. \tag{1}$$

**Distribution Matching Distillation (DMD)**. To distill a pretrained multi-step diffusion model into a single-step generator $G_\theta$, a distribution matching approach can be used [64, 65]. This method seeks to minimize the expected approximate Kullback–Leibler (KL) divergence, $\mathcal{L}_{\text{DMD}} = \mathbb{E}_t(\mathbb{D}_{\text{KL}}(p_{\text{fake},t} \| p_{\text{real},t}))$, between the diffused target distribution, $p_{\text{real},t}(\mathbf{x})$, and the diffused generator distribution, $p_{\text{fake},t}$, across different timesteps $t$ with the following objective function:

$$\nabla_\theta \mathcal{L}_{\text{DMD}} = -\mathbb{E}_t \left( \int (s_{\text{real}}(F(G_\theta(\mathbf{z}), t), t) - s_{\text{fake}}(F(G_\theta(\mathbf{z}), t), t)) \frac{dG_\theta(\mathbf{z})}{d\theta} d\mathbf{z} \right), \tag{2}$$

where $\mathbf{z} \sim \mathcal{N}(0, \mathbf{I})$ is the guassian noise, $F$ is the diffusion forward process with noise level corresponding to time step $t$ and $s_{\text{real}}$ and $s_{\text{fake}}$ are score function defined in Equation (1). DMD employs a frozen pre-trained diffusion model, $\mu_{\text{real}}$ as the teacher, while iteratively updating $\mu_{\text{fake}}$ during the training of $G_\theta$. This involves a denoising score-matching loss applied to samples produced by the one-step generator $G_\theta(\mathbf{z})$, where $\mathbf{z} \sim \mathcal{N}(0, \mathbf{I})$ is a guassian noise.

**Reinforcement Learning.** When fine-tuning models with reinforcement learning, the learned reward function serves as a feedback to guide the model's training. Specifically, given a noise vector $\mathbf{z} \sim \mathcal{N}(0, \mathbf{I})$, we generate an image based on this input. Following a widely adopted optimization objective from the fine-tuning of large language models (LLMs) [20, 21], the training objective is:

$$\max_\theta \mathbb{E}_{\mathbf{z} \sim \mathcal{N}(0,\mathbf{I}), \mathbf{x} \sim G_\theta(\mathbf{x}|\mathbf{z})} [r_\phi(\mathbf{x}, \mathbf{z})] - \beta \mathbb{D}_{\text{KL}} [G_\theta(\mathbf{x} \mid \mathbf{z}) \| G_{\theta'}(\mathbf{x} \mid \mathbf{z})], \tag{3}$$

where we consider modeling the reinforcement learning process for the one-step generator $G_\theta$ and $\theta'$ serves as a reference model from a previous optimization step or a supervised fine-tuned model, similar to those commonly used in large language model literature. The reward function $r_\phi(\cdot)$ is trained prior to the reinforcement learning stage and provides feedback as the reward model.

# 4 Simple Distillation Approach for One-Step Diffusion Models

In this section, we introduce a simple Contrastive Energy Distillation (CED) based approach to training a one-step generator that mimics the generative process of a multistep diffusion model. We introduce a joint energy-based model that represents the auxiliary (fake) score model $s_{\text{fake}}$ and the one-step generator $G_\theta$. To simplify the training pipeline and avoid iterative updates involving auxiliary models (fake score models), we employ CED to directly train the one-step generator using an approach inspired by Noise Contrastive Estimation (NCE). We leverage a connection between CED and distribution matching distillation methods to design a simple and efficient method to distill a multi-step diffusion model into a one-step generator. Our approach requires loading the student model only during training, significantly reducing computational complexity and memory use while achieving competitive performance relative to state-of-the-art methods.

## 4.1 Joint Energy-based Models

We can interpret the two-step learning process of one-step diffusion models as leveraging a "fake score model," $s_{\text{fake}}$, designed to approximate the distribution of the one-step generator. However, during iterative training, optimization errors may prevent $s_{\text{fake}}$ from accurately reflecting the true distribution of the generator. Conceptually, this discrepancy can be viewed as an energy function that captures the difference between samples generated by the fake score model and those from the one-step generator. Based on this viewpoint, we introduce an unnormalized density to simultaneously represent the fake score model's distribution and its deviation from the one-step generator.

$$p_{\theta,\phi}\left(\mathbf{x} \mid \mathbf{z}\right) = p_{\text{fake},\phi}\left(\mathbf{x} \mid \mathbf{z}\right) \frac{\exp\left(-\boldsymbol{E}\left(\mathbf{x}, G_\theta(\mathbf{z})\right)\right)}{Z_\phi\left(\mathbf{z}\right)}, \tag{4}$$

where $Z_\phi$ is the partition function as a normalizing factor. The term $p_{\theta,\phi}\left(\mathbf{x} \mid \mathbf{z}\right)$ denotes the distribution that generates samples $\mathbf{x}$ from an input $\mathbf{z}$ drawn from pure Gaussian noise $\mathbf{z} \sim \mathcal{N}(0, \mathbf{I})$. Similarly, $p_{\text{fake},\theta}\left(\mathbf{x} \mid \mathbf{z}\right)$ denotes the distribution of samples produced by iteratively denoising $\mathbf{z} \sim \mathcal{N}(0, \mathbf{I})$ by the reverse process of diffusion model based on the fake score model $s_{\text{fake},\theta}$ and $\mathbf{x}$. $\boldsymbol{E}$ is used to measure the discrepancy between generated samples from fake score models and one-step generators. Moreover, $\boldsymbol{E}(\mathbf{x}, G_\theta(\mathbf{z}))$ can be viewed as a distance function between samples produced by the fake score models and those generated by the one-step generator—using, for instance, the L2, Pseudo-Huber, or LPIPS-Huber distance [6]. In this formulation, a higher energy value indicates a larger discrepancy between the two sets of samples, while a lower energy implies a lower discrepancy. Therefore, this joint EBM provides a calibrated distribution for $p_{\text{fake}}$: samples that are well-aligned with the one-step generator are assigned higher probabilities, while poorly aligned samples receive lower probabilities. In other words, $p_{\theta,\phi}$ offers a more accurate estimation of the image distribution of $G_\theta$, and also capturing the interplay between the fake score model and the one-step generator.

## 4.2 Training with Contrastive Energy Distillation

The goal of distillation methods is to align the distribution of the fake score model ($p_{\text{fake}}$) with that of the teacher diffusion model ($p_{\text{real}}$). In this paper, we use the calibrated distribution (joint EBM) $p_{\theta,\phi}$ introduced in earlier sections, to approximate the teacher model $p_{\text{real}}$. Accordingly, we treat the joint EBM defined in Equation (4) as modeling the teacher distribution $p_{\text{real}}$. Based on this formulation, we optimize the parameters of the joint energy function using a conditional variant of Noise Contrastive Estimation (NCE) [16]. First, NCE samples contrastive examples from both the data distribution and a noise distribution that should closely approximate the data distribution. Second, it computes the likelihoods of these samples under both the model and the noise distributions, then optimizing a binary classification objective. Under our joint energy formulation from Equation (1), the log-odds reduce to $\log p_{\theta,\phi} - \log p_\phi = -\boldsymbol{E}(\mathbf{x}, G_\theta(\mathbf{z}))$, simplifying the objective to a standard binary classification form with $\mathbf{z} \sim \mathcal{N}(0, \mathbf{I})$ and referred to Contrastive Energy Distillation (CED):

$$\mathcal{L}_{\text{CED}} = -\mathbb{E}_{\mathbf{x} \sim p_{\theta,\phi}}[\log \frac{1}{1 + \exp\left(\boldsymbol{E}\left(\mathbf{x}, G_\theta(\mathbf{z})\right)\right)}] - \mathbb{E}_{\mathbf{x}' \sim p_\phi}[\log \frac{1}{1 + \exp\left(-\boldsymbol{E}\left(\mathbf{x}', G_\theta(\mathbf{z})\right)\right)}]. \tag{5}$$

In this formulation, we replace $p_{\theta,\phi}$ with the teacher model $p_{\text{real}}(\mathbf{x} \mid \mathbf{z})$ as our target distilled distribution for sampling positive examples in the first term. Meanwhile, one-step diffusion distillation typically involves fitting the fake score model $p_\phi$ to outputs from the generator $G_{\theta'}(\mathbf{z})$ of the previous

---

**Algorithm 1** Simple Distillation of One-step Diffusion Models

---
**Require:** Teacher diffusion model $p_{\text{true}}$
1: Initialize the one-step generator with the teacher's score network $G_{\theta_{\text{init}}}(\cdot) \equiv s_{\text{true}}(\cdot)$
2: Sample $\mathbf{x}$ from $p_{\text{true}}(\mathbf{x} \mid \mathbf{z})$ with $\mathbf{z} \sim \mathcal{N}(0, \mathbf{I})$
3: Warm-up train with $\mathcal{L} = \boldsymbol{E}(\mathbf{x}, \mathbf{z})$ to obtain a ones-step generator $G_{\theta'}$
4: Get negative samples $\mathbf{x}'$ from $G_{\theta'}(\mathbf{z})$ and construct a triplet $(\mathbf{x}, \mathbf{x}', \mathbf{z})$ for each $\mathbf{z} \sim \mathcal{N}(0, \mathbf{I})$
5: **for** each training iteration **do**
6:     Get a batch of samples $(\mathbf{x}, \mathbf{x}', \mathbf{z})$
7:     Update one-step generator's parameters $\theta$ with $\mathcal{L}_{\text{CED}}$ in Equation (14) with $(\mathbf{x}, \mathbf{x}', \mathbf{z})$
8: **end for**

---

optimization step, where $\theta'$ denotes the earlier parameter set of $\theta$. To streamline the iterative process and avoid training an additional fake-score model, we replace $p_\phi$ with $G_{\theta'}$ for the negative samples in the second term of Equation (5). Since $p_\phi$ is designed to approximate the distribution of images from the previous optimization step—following earlier iterative learning approaches for one-step diffusion models—this substitution remains valid. Thus, we propose the following loss to distill the teacher model without extra models or iterative training, thereby simplifying the distillation process:

$$\mathcal{L}_{\text{CED}} = \mathbb{E}_{\mathbf{x} \sim p_{\text{real}}}[\log\left(1 + \exp\left(\boldsymbol{E}\left(\mathbf{x}, G_\theta(\mathbf{z})\right)\right)\right)] + \mathbb{E}_{\mathbf{x}' \sim G_{\theta'}}[\log\left(1 + \exp\left(-\boldsymbol{E}\left(\mathbf{x}', G_\theta(\mathbf{z})\right)\right)\right)], \quad (6)$$

where the energy function $\boldsymbol{E}$ serves as a distance metric, where smaller values correspond to reduced discrepancy between the two input samples. For instance, when using the L2 distance, we have $\boldsymbol{E}(\mathbf{x}, G_\theta(\mathbf{z})) = \|x - G_\theta(\mathbf{z})\|_2^2$. To further streamline the training process, we begin with a brief warm-up phase for the one-step generator $G_{\theta'}$. We then optimize the objective specified in Equation (6). The full details of this procedure can be found in Algorithm 1.

### 4.3 Relation to Distribution Matching Distillation

In this section, we establish the connection between our proposed objective and Distribution Matching Distillation (DMD) methods. We begin by formulating the KL divergence objective of distribution matching distillation from a reinforcement learning perspective:

$$\max_\theta -\mathbb{D}_{\text{KL}}\left(p_{\text{fake}}^\theta(\mathbf{x} \mid \mathbf{z}) \| p_{\text{real}}(\mathbf{x} \mid \mathbf{z})\right) = \mathbb{E}_{p_{\text{fake}}^\theta}\left[\log p_{\text{real}}(\mathbf{x} \mid \mathbf{z}) - \log p_{\text{fake}}(\mathbf{x} \mid \mathbf{z})\right]$$

$$\mathbb{E}_{p_{\text{fake}}^\theta}\left[\log \frac{p_{\text{real}}(\mathbf{x} \mid \mathbf{z})}{p_{\text{fake}}^{\theta_0}(\mathbf{x} \mid \mathbf{z})} - \log \frac{p_{\text{fake}}^\theta(\mathbf{x} \mid \mathbf{z})}{p_{\text{fake}}^{\theta_0}(\mathbf{x} \mid \mathbf{z})}\right] = \mathbb{E}_{p_{\text{fake}}^\theta} r(\mathbf{x}, \mathbf{z}) - \mathbb{D}_{\text{KL}}\left(p_{\text{fake}}^\theta(\mathbf{x} \mid \mathbf{z}) \| p_{\text{fake}}^{\theta_0}(\mathbf{x} \mid \mathbf{z})\right), \quad (7)$$

where $r(\mathbf{x}, \mathbf{z}) = \log(p_{\text{real}}(\mathbf{x} \mid \mathbf{z})/p_{\text{fake}}^{\theta_0}(\mathbf{x} \mid \mathbf{z}))$ can be viewed as an auxiliary reward function and $\theta_0$ is the initial parameter of the fake score model, initialized from the teacher diffusion model. Hence, DMD can be interpreted in a manner analogous to Reinforcement Learning From Human Feedback (RLHF) [39, 41, 60, 58, 59]. Specifically, the procedure has two stages: (1) learning a reward model $r$, and (2) optimizing the primary objective. To relate DMD to our single-step training approach, we reformulate it so that these two steps are combined into a single unified procedure. Specifically, we begin with the standard RLHF training pipeline, where the first step is to learn a reward model. For the DMD objective in Equation (7), we use Density Ratio Estimation methods to learn the reward model $\log(p_{\text{real}}(\mathbf{x} \mid \mathbf{z})/p_{\text{fake}}^{\theta_0}(\mathbf{x} \mid \mathbf{z}))$. Since this process relies on estimating the log ratio, a straightforward solution is to train a classifier (i.e., a discriminator) with logistic regression to approximate it:

$$\mathcal{L}_{\text{DRE}} = -\mathbb{E}_{\mathbf{x} \sim p_{\text{real}}}[\log\left(\sigma(h(\mathbf{x}, \mathbf{z}))\right)] - \mathbb{E}_{\mathbf{x}' \sim p_{\text{fake}}^{\theta_0}}[\log\left(1 - \sigma(h(\mathbf{x}, \mathbf{z}))\right)], \quad (8)$$

where $\sigma(\cdot)$ denotes the sigmoid function, and each data sample is treated as though it is drawn from a distribution with binary labels—one class for samples from $p_{\text{real}}$ and one class for samples from $p_{\text{fake}}^{\theta_0}$. Then, the log density ratio can be linked to the optimal classifier probabilities via Bayes' rule [2]:

$$\log \frac{p_{\text{real}}(\mathbf{x} \mid \mathbf{z})}{p_{\text{fake}}^{\theta_0}(\mathbf{x} \mid \mathbf{z})} = \frac{P(c=1)P(c=0 \mid \mathbf{x}, \mathbf{z})}{P(c=0)P(c=1 \mid \mathbf{x}, \mathbf{z})} = \log\left(\frac{\sigma\left(h^*(\mathbf{x}, \mathbf{z})\right)}{1 - \sigma\left(h^*(\mathbf{x}, \mathbf{z})\right)}\right), \quad (9)$$

where $h^*(\cdot)$ is the optimal solution for Equation (8) and is the constant ratio $P(c=1)/P(c=0)$ between the priors of two classes that can be estimated with sample size. Although we can optimize

Equation (7) with the learned reward (i.e., density ratio) from Equation (8), this still involves a two-step process and additional computational overhead. To simplify matters and align with our proposed objective of Equation (6), we propose a direct optimization method for the DMD objective that does not require an RL training loop or a separate discriminator. The key insight lies in using a specific parameterization for the discriminator, which allows us to extract the optimal solution in closed form—circumventing the iterative RL loop in Equation (7). Specifically, under this parameterization, the optimization problem defined in Equation (7) admits a straightforward analytical solution:

$$p_{\text{fake}}^{\theta^*}(\mathbf{x} \mid \mathbf{z}) = \frac{1}{Z(\mathbf{z})} p_{\text{fake}}^{\theta_0}(\mathbf{x} \mid \mathbf{z}) \exp(r(\mathbf{x}, \mathbf{z})), \tag{10}$$

where $Z(\mathbf{z}) = \int_{\mathbf{x}} p_{\text{fake}}^{\theta_0}(\mathbf{x} \mid \mathbf{z}) \exp(r(\mathbf{x}, \mathbf{z})) = \int_{\mathbf{x}} p_{\text{real}}(\mathbf{x} \mid \mathbf{z}) = 1$ by the definition of $r(\cdot)$ in Equation (7) and $\theta^*$ is the optimal solution of paramter $\theta$. A detailed derivation of this optimal format (Equation (10)) is provided in the Appendix A.1. To unify the process into a single optimization step, we can combine Equations (10) and (9) with simple algebra, yielding the following relationship:

$$\log \frac{p_{\text{fake}}^{\theta^*}(\mathbf{x} \mid \mathbf{z})}{p_{\text{fake}}^{\theta_0}(\mathbf{x} \mid \mathbf{z})} = \log \left( \frac{\sigma\left(h^*(\mathbf{x}, \mathbf{z})\right)}{1 - \sigma\left(h^*(\mathbf{x}, \mathbf{z})\right)} \right) \Rightarrow h^*(\mathbf{x}, \mathbf{z}) = \log \frac{p_{\text{fake}}^{\theta^*}(\mathbf{x} \mid \mathbf{z})}{p_{\text{fake}}^{\theta_0}(\mathbf{x} \mid \mathbf{z})}. \tag{11}$$

Then, by combining the two-step optimization in Equation (8) with Equations (7) and (5), we derive a direct optimization algorithm, resulting in the following objective:

$$\mathcal{L}_{\text{DRE}}^* = -\mathbb{E}_{\mathbf{x} \sim p_{\text{real}}} \left[ \log \left( \sigma(\log \frac{p_{\text{fake}}^{\theta^*}(\mathbf{x} \mid \mathbf{z})}{p_{\text{fake}}^{\theta_0}(\mathbf{x} \mid \mathbf{z})}) \right) \right] - \mathbb{E}_{\mathbf{x}' \sim p_{\text{fake}}^{\theta_0}} \left[ \log \left( 1 - \sigma(\log \frac{p_{\text{fake}}^{\theta^*}(\mathbf{x} \mid \mathbf{z})}{p_{\text{fake}}^{\theta_0}(\mathbf{x} \mid \mathbf{z})}) \right) \right]. \tag{12}$$

For $p_{\text{fake}}^{\theta^*}(\mathbf{x} \mid \mathbf{z})$, we can use a single-step generator parameter, modeling the generator as a Gaussian distribution $p_{\text{fake}}^{\theta} \sim \mathcal{N}(G_\theta, \sigma^2 \mathbf{I})$. Then, we can convert this objective with the trainable parameters by replacing $\theta^*$ with $\theta$ and we can see the relation with our proposed objective in Equation (6):

$$\mathcal{L}_{\text{DRE}} = \mathbb{E}_{\mathbf{x} \sim p_{\text{real}}}[\log\left(1 + \exp\left(\boldsymbol{E}\left(\mathbf{x}, G_\theta(\mathbf{z})\right)\right)\right)] + \mathbb{E}_{\mathbf{x}' \sim G_{\theta'}}[\log\left(1 + \exp\left(-\boldsymbol{E}\left(\mathbf{x}', G_\theta(\mathbf{z})\right)\right)\right)], \tag{13}$$

where $\boldsymbol{E}\left(\mathbf{x}, G_\theta(\mathbf{z})\right) = -(\log p_{\text{fake}}^{\theta}(\mathbf{x} \mid \mathbf{z}) - \log p_{\text{fake}}^{\theta_0}(\mathbf{x} \mid \mathbf{z})) \approx -(\|\mathbf{x} - G_\theta(\mathbf{z})\|_2^2 - \|\mathbf{x} - G_{\theta'}(\mathbf{z})\|_2^2)$. Therefore, although the objective in Equation (8) requires a two-step training process from the reinforcement learning perspective, we unify these steps by noting that they have the relation between two steps' optimal parameter (see Equations (11) and (12)). By optimizing Equation (13) to its optimal solution, we implicitly fulfill the DMD objective, effectively merging the original two-step learning procedure into a single step. This streamlined strategy aligns closely with methods commonly employed in Direct Alignment Algorithms for RLHF [41], which we further discuss in Appendix A.2. Furthermore, implicitly optimizing DMD objectives can be viewed as a special case of our proposed CED framework, realized through a specific definition of the underlying energy functions.

## 4.4 Generalized Extensions and Practical Implementations

We have presented a simple distillation algorithm for one-step diffusion models that aligns with distribution-matching objectives in Equation (7), and we will next explore an extension of this approach using different loss formulations. Since our CED in Equation (13) and (6) can be seen as a binary classification problem, it naturally extends to a more general form [5, 13]:

$$\mathcal{L}_{\text{CED}} = \mathbb{E}_{\mathbf{x} \sim p_{\text{real}}} f^+(-\boldsymbol{E}(\mathbf{x}, G_\theta(\mathbf{z}))) + \mathbb{E}_{\mathbf{x}' \sim G_{\theta'}} f^-(-\boldsymbol{E}(\mathbf{x}', G_\theta(\mathbf{z}))). \tag{14}$$

We summarize the different classification loss functions $f^+$ and $f^-$ (e.g., Logistic, Hinge [7], Brier [13], Exponential [12]) in Table 1, including the Logistic loss from Equation (11). For practical implementation, the complete training procedure is detailed in Algorithm 1. To simplify the training process and reduce computational costs, we first generate synthetic samples from the teacher diffusion model and perform a warm-up phase by optimizing the one-step generator to reconstruct images from pure Gaussian noise. Specifically, we minimize the discrepancy

Table 1: Summary of various functions of generalized extensions of CED for Equation (14).

| Loss | $f^+(-\boldsymbol{E})$ | $f^-(-\boldsymbol{E})$ |
|---|---|---|
| Logistic | $\log(1 + e^{\boldsymbol{E}})$ | $\log(1 + e^{-\boldsymbol{E}})$ |
| Hinge | $\max(0, 1 + \boldsymbol{E})$ | $\max(0, 1 - \boldsymbol{E})$ |
| Brier | $\left(\frac{e^{\boldsymbol{E}}}{1+e^{\boldsymbol{E}}}\right)^2$ | $\left(\frac{1}{1+e^{\boldsymbol{E}}}\right)^2$ |
| Exponential | $e^{\boldsymbol{E}}/2$ | $e^{-\boldsymbol{E}}/2$ |

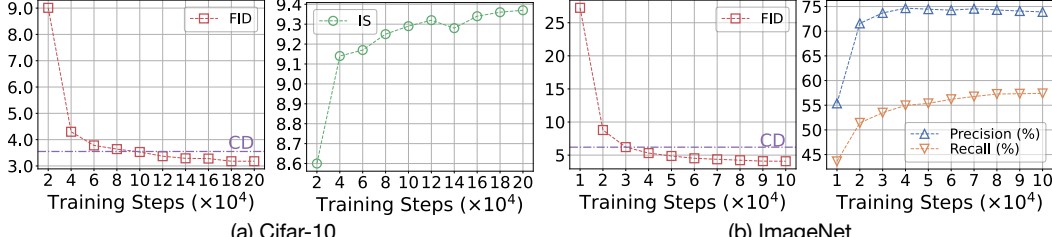

Figure 2: Results on Cifar-10 and ImageNet of different training steps, where CD [52] requires 800,000 steps to reach FID 3.55 on CIFAR-10 and 600,000 steps to get FID 6.20 on ImageNet.

with the auxiliary energy function $\boldsymbol{E}(\mathbf{z})$, obtaining an intermediate model $\theta'$, which subsequently provides negative samples for the final training objective (Line 3). To further enhance efficiency, we construct a dataset consisting of triplets $(\mathbf{x}, \mathbf{x}', \mathbf{z})$ where each noise vector $\mathbf{z}$ corresponds to positive samples $\mathbf{x}$ from the teacher diffusion model and negative samples $\mathbf{x}'$ from the warm-up generator $G_{\theta'}$ before training. This dataset with $(\mathbf{x}, \mathbf{x}', \mathbf{z})$ is then used to optimize $G_\theta$ according to Equation (14).

## 5 Experiment

### 5.1 Experimental Setup

**Datasets and Models.** To thoroughly evaluate CED, we use two representative benchmark datasets from prior works: CIFAR-10 (32×32) for unconditional generation [28] and ImageNet (64×64) for conditional generation [8]. To distill the teacher diffusion models, we employ EDM models [22], which are widely used in distillation-based methods.

**Evaluation Protocol.** We measure image generation quality using the Frechet Inception Distance (FID) and the Inception Score (IS) [38]. For FID, we generate 50k samples and compare them against the training set used by the EDM teacher model as the reference. We also consider Precision and Recall when evaluating conditional generation on ImageNet 64x64, where we follow previous works with a predefined reference batch to compute both metrics [9, 52].

**Baselines.** We assess the effectiveness of one-step distillation methods and other efficient diffusion models without directly utilizing the original image dataset, instead focusing on comparing the quality of images generated by CED. Based on the models employed during student model training, we categorize the baseline methods into the following groups: (1) Diffusion Models, which include methods aimed at accelerating sampling speed as well as original diffusion models; (2) (Generative Adversarial Networks) GANs with extra discriminator models; (3) Distillation with Teacher & Fake Score Models, which utilize both Teacher and Auxiliary (Fake) Score Models; (4) Distillation with Teacher Models; and (5) Distillation with Student Models Only. Note that these categorizations are based specifically on the models involved during the training of student generators.

**Implementation Details.** We implement CED on top of the EDM codebase [22], initializing the one-step generator $\theta$ with $\theta_{\text{true}}$ from the teacher diffusion EDM model. To reduce computational overhead in Equation (14) (see Algorithm 1), we randomly sample negative samples for only 25% of the batch size—compared to the positive samples—to optimize the objective effectively. Moreover, we compute the energy function using the LPIPS-Huber distance [51, 29], which balances perceptual similarity and robustness to outliers. As outlined in Algorithm 1, we first sample from the teacher model to generate training data. Specifically, we use 35 sampling steps for the Cifar-10 and 79 steps for ImageNet with Heun's 2nd-order method [22], as shown in Line 2 of Algorithm 1.

### 5.2 Image Generation

We present the image generation results comparing various baseline methods on the CIFAR-10 and ImageNet datasets in Tables 2 and 3, respectively. Specifically, we include two variants of our approach: CED-DRE and CED. The key difference between these variants is whether we load the previously obtained warm-up student model parameters $\theta_0$ during training. From the density ratio estimation perspective, CED-DRE calculates the energy as $\boldsymbol{E}(\mathbf{x}, G_\theta(\mathbf{z})) = -(\|\mathbf{x} - G_\theta(\mathbf{z})\|_2^2 - \|\mathbf{x} - G_{\theta'}(\mathbf{z})\|_2^2)$, where we replace the traditional L2 distance with the LPIPS-Huber distance. Firstly, we observe that preserving the intermediate model parameters $\theta'$ does not significantly impact the final performance. Therefore, to further save memory, we can safely omit storing these parameters. Instead,

Table 2: Results of unconditional image generation on CIFAR-10 with FID and IS.

| METHOD | NFE (↓) | FID (↓) | IS (↑) |
|---|---|---|---|
| **Diffusion models** | | | |
| Score SDE [53] | 2000 | 2.38 | 9.83 |
| DDPM [18] | 1000 | 3.17 | 9.46 |
| LSGM [54] | 147 | 2.10 | |
| EDM [22] | 35 | 1.97 | |
| **GANs** | | | |
| BigGAN [4] | 1 | 14.7 | 9.22 |
| StyleGAN2 [24] | 1 | 8.32 | 9.21 |
| StyleGAN2-ADA [23] | 1 | 2.92 | 9.83 |
| **Distillation with Teacher & Fake Score Models** | | | |
| Diff-Instruct [35] | 1 | 4.53 | 9.89 |
| DMD [65] | 1 | 3.77 | |
| SID [70] | 1 | 1.92 | 9.98 |
| **Distillation with Teacher Models** | | | |
| Knowledge Distillation [34] | 1 | 9.36 | |
| DFNO (LPIPS) [68] | 1 | 3.78 | |
| TRACT [1] | 1 | 3.78 | |
| PD [46] | 1 | 9.12 | |
| CD (LPIPS) [52] | 1 | 3.55 | 9.48 |
| CTM w/o GAN [25] | 1 | 5.19 | |
| 1-rectified flow (+distill) [31] | 1 | 6.18 | 9.08 |
| 2-rectified flow [31] | 1 | 12.21 | 8.08 |
| +distill [31] | 1 | 4.85 | 9.01 |
| 3-rectified flow [31] | 1 | 8.15 | 8.47 |
| +Distill [31] | 1 | 5.21 | 8.79 |
| **Distillation with Student Models Only** | | | |
| CED-DRE | 1 | **2.95** | 9.36 |
| CED | 1 | 2.96 | **9.42** |

Table 3: Results of conditional image generation on ImageNet 64 × 64 with FID, Precision and Recall.

| METHOD | NFE (↓) | FID (↓) | Prec. (↑) | Rec. (↑) |
|---|---|---|---|---|
| **Diffusion models** | | | | |
| DDIM [50] | 50 | 13.7 | 0.65 | 0.56 |
| | 10 | 18.3 | 0.60 | 0.49 |
| DPM solver [33] | 10 | 7.93 | | |
| | 20 | 3.42 | | |
| DEIS [66] | 10 | 6.65 | | |
| | 20 | 3.10 | | |
| DDPM [18] | 250 | 11.0 | 0.67 | 0.58 |
| iDDPM [38] | 250 | 2.92 | 0.74 | 0.62 |
| ADM [9] | 250 | 2.07 | 0.74 | 0.63 |
| EDM [22] | 79 | 2.30 | | |
| **GANs** | | | | |
| BigGAN-deep [4] | 1 | 4.06 | 0.79 | 0.48 |
| **Distillation with Teacher & Fake Score Models** | | | | |
| Diff-Instruct [35] | 1 | 5.57 | | |
| DMD [65] | 1 | 2.62 | | |
| SID [70] | 1 | 1.52 | 0.74 | 0.63 |
| **Distillation with Teacher Models** | | | | |
| TRACT [1] | 1 | 7.43 | | |
| BOOT [15] | 1 | 16.3 | 0.68 | 0.36 |
| PD [46] | 2 | 15.39 | 0.59 | 0.62 |
| CD (LPIPS) [52] | 1 | 6.20 | 0.68 | 0.6 |
| CTM w/o GAN [25] | 2 | 5.8 | | |
| **Distillation with Student Models Only** | | | | |
| CED-DRE | 1 | **3.88** | 0.72 | **0.60** |
| CED | 1 | 4.06 | **0.74** | 0.58 |

we pre-sample negative examples and directly load the current student model itself during training, as outlined in Algorithm 1. Furthermore, our results consistently demonstrate that CED outperforms baseline methods using only teacher models like Consistency Models (CM) and Flow Matching (FM), while significantly reducing memory consumption of loading models by approximately 50% during training. Also, our method CED can outperform DMD which includes iterative training with extra models on CIFAR-10 datasets. Although our method does not surpass approaches involving both teacher and auxiliary (fake) score models with iterative training like DMD and SiD on ImageNet, it simplifies the training pipeline by eliminating the necessity of loading and maintaining three models simultaneously, thus achieving a 66% reduction in memory usage of loading models. Also, unlike these methods—which require iterative updates and thus demand careful hyperparameter tuning for multiple models—CED avoids extra hyperparameter search, streamlining the overall training process.

Moreover, we evaluate CED at different training steps on CIFAR-10 and ImageNet, as shown in Figure 2. Remarkably, CED achieves strong performance with just 100,000 training steps on CIFAR-10 (FID 3.53) and 40,000 steps on ImageNet (FID 5.31). In comparison, CD requires 800,000 steps to reach a similar FID of 3.55 on CIFAR-10 and 600,000 steps to achieve an FID of 6.20 on ImageNet, while also requiring the loading of additional models during training (based on their official code implementation). These results further demonstrate the efficiency of our proposed algorithm, which achieves good performance with relatively few training steps. Notably, CED does not incur additional training overhead. This efficiency likely stems from the fact that CED avoids the need for extra timestep sampling like previous distillation methods (e.g., CD) during training, thereby simplifying the process and potentially enabling faster convergence.

## 5.3 Analysis and Ablation Study

In this section, we conduct experiments to analyze the impact of various distance metrics used in the energy function and different loss formulations together with analysis of generated imge quality, as summarized in Table 1.

**Ablation Study with Different Distance Functions**. We investigate the impact of different distance functions on

Table 4: Results on Cifar-10 with various distance functions for the energy $E(\cdot)$.

| Method | FID (↓) | IS (↑) |
|---|---|---|
| L2 | 4.07 | 9.23 |
| L1 | 3.78 | 9.23 |
| Pseudo-Huber | 3.04 | 9.34 |
| LPIPS-Huber | **2.96** | **9.42** |

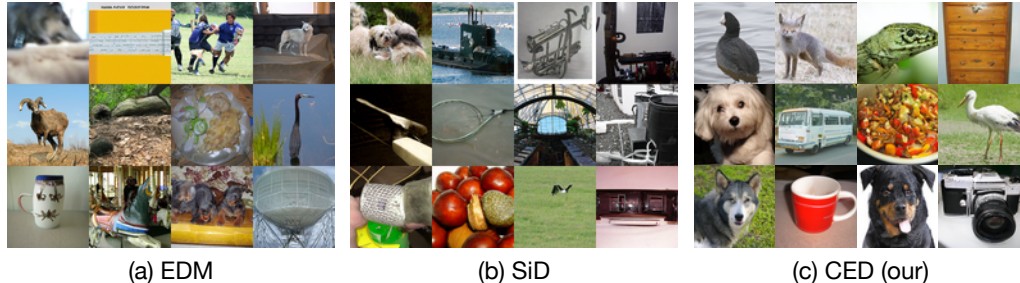

| (a) EDM | (b) SiD | (c) CED (our) |

Figure 4: Case Study of generated images from different models trained on ImageNet.

model training. Specifically, we modify the energy function $E(\mathbf{x}, G_\theta(\mathbf{z}))$ in Equation (14) by substituting various loss functions, including L2 distance, L1 distance, Pseudo-Huber [6, 51], and LPIPS-Huber [29]. We adopt the logistic loss formulation as described in Table 1, and the corresponding results on CIFAR-10 are reported in Table 4. The Pseudo-Huber distance function, known for being less sensitive to outliers than the squared L2 distance, can potentially reduce gradient variance during training. Our results show that it consistently outperforms the standard L2 distance on Cifar-10, indicating its effectiveness. We also explore the LPIPS-Huber distance function, which encourages the model to minimize perceptual differences between generated samples and the ground truth, as demonstrated in prior works on generative modeling [29]. Among the tested metrics, the Pseudo-Huber distance function significantly outperforms L2 and other alternatives, as summarized in Table 4. Therefore, we select the LPIPS-Huber distance as the primary choice for our model.

**Ablation Study with Different Loss Functions**. We investigate the impact of different loss function formulations on the final objective in Equation 14. The specific formats of these loss functions are presented in Table 1. Our results show that both the Logistic and Hinge loss formulations achieve the best performance compared to other alternatives, making them strong candidates for practical use for training a one-step diffusion model. Additionally, in our implementation, we observe that Hinge loss leads to faster convergence (i.e., fewer training steps) compared to Logistic loss, making it a particularly attractive choice when aiming for competitive results with reduced training time.

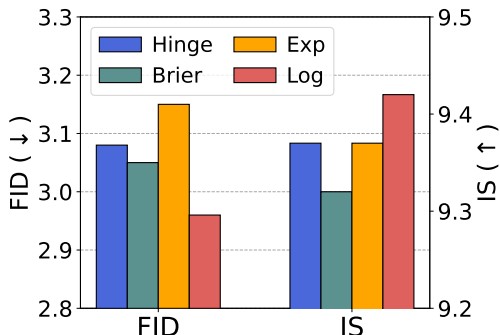

Figure 3: Results on CIFAR-10 with different loss functions, as shown in Table 1. "Exp" denotes Exponential and "Log" denotes Logistic.

**Analysis of Generated Image Quality**. We further generate images using different models, including EDM (our teacher model), SiD (a state-of-the-art one-step generator) and our proposed method (CED). We selected representative images from each model for qualitative comparison. Although our model (CED) does not outperform SiD in terms of FID scores, it achieves comparable image quality, as illustrated in Figure 4. Moreover, the quality of images generated by CED are also comparable to those produced by the teacher model, further demonstrating the effectiveness of our approach to generate high quality images. Notably, CED requires only one third of the memory to load models during training, making it particularly attractive for scaling to larger models with more parameters. More examples CED-generated images are included in the Appendix C.

## 6   Conclusion

In conclusion, we introduce CED, a simple and efficient distillation algorithm leveraging noise contrastive estimation to distill multi-step diffusion models into a single-step generator. Our method implicitly optimizes the KL divergence commonly used for distribution-matching distillation, providing theoretical justification for its effectiveness. Empirically, CED outperforms existing one-step training methods that avoid iterative procedures but still require loading the teacher model during training, incurring additional memory overhead. In contrast, CED significantly simplifies the pipeline for one-step image generation while achieving superior performance. We believe this work paves the way for a new class of one-step distillation techniques and holds strong potential for extension to larger-scale tasks, such as text-to-image generation with high-capacity models.

## Acknowledge

This work was supported in part by grants from the National Science Foundation (2226025) and the National Center for Advancing Translational Sciences and the National Institutes of Health (UL1 TR002014) and by the Center for Artificial Intelligence Foundations and Scientific Applications and the Institute for Computational and Data Sciences at Pennsylvania State University.

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

## A  Omitted Derivation Details and Background

### A.1  Derivation of Equation (10)

We derive Equation (9) following [41, 57]. We first consider the Distribution Matching Distillation objective in Equation (7) from the reinforcement learning perspective:

$$\max_\theta \mathbb{E}_{p_{\text{fake}}^\theta} r(\mathbf{x}, \mathbf{z}) - \beta \mathbb{D}_{\text{KL}} \left[ p_{\text{fake}}^\theta(\mathbf{x} \mid \mathbf{z}) \| p_{\text{fake}}^{\theta_0}(\mathbf{x} \mid \mathbf{z}) \right], \tag{15}$$

where we introduce a hyperparameter $\beta$ to control the weight of KL divergence. The reward function is $r(\mathbf{x}, \mathbf{z}) = \log(p_{\text{real}}(\mathbf{x} \mid \mathbf{z})/p_{\text{fake}}^{\theta_0}(\mathbf{x} \mid \mathbf{z}))$ and $\theta_0$ is the parameter of the reference model. Then, we have:

$$
\begin{aligned}
&\max_\theta \mathbb{E}_{\mathbf{z} \sim \mathcal{N}(0,\mathbf{I}), \mathbf{x} \sim p_{\text{fake}}^\theta}[r(\mathbf{x}, \mathbf{z})] - \beta \mathbb{D}_{\text{KL}} \left[ p_{\text{fake}}^\theta(\mathbf{x} \mid \mathbf{z}) \| p_{\text{fake}}^{\theta_0}(\mathbf{x} \mid \mathbf{z}) \right] \\
&= \max_\theta \mathbb{E}_{\mathbf{z} \sim \mathcal{N}(0,\mathbf{I})} \mathbb{E}_{p_{\text{fake}}^\theta} \left[ r(\mathbf{x}, \mathbf{z}) - \beta \log \frac{p_{\text{fake}}^\theta(\mathbf{x} \mid \mathbf{z})}{p_{\text{fake}}^{\theta_0}(\mathbf{x} \mid \mathbf{z})} \right] \\
&= \min_\theta \mathbb{E}_{\mathbf{z} \sim \mathcal{N}(0,\mathbf{I})} \mathbb{E}_{p_{\text{fake}}^\theta} \left[ \log \frac{p_{\text{fake}}^\theta(\mathbf{x} \mid \mathbf{z})}{p_{\text{fake}}^{\theta_0}(\mathbf{x} \mid \mathbf{z})} - \frac{1}{\beta} r(\mathbf{x}, \mathbf{z}) \right] \\
&= \min_\theta \mathbb{E}_{\mathbf{z} \sim \mathcal{N}(0,\mathbf{I})} \mathbb{E}_{p_{\text{fake}}^\theta} \left[ \log \frac{p_{\text{fake}}^\theta(\mathbf{x} \mid \mathbf{z})}{\frac{1}{Z(\mathbf{z})} p_{\text{fake}}^{\theta_0}(\mathbf{x} \mid \mathbf{z}) \exp\left(\frac{1}{\beta} r(\mathbf{x}, \mathbf{z})\right)} - \log Z(\mathbf{z}) \right],
\end{aligned}
\tag{16}
$$

where $Z(\mathbf{z}) = \sum_{\mathbf{x}} \pi_{\text{ref}}(\mathbf{x} \mid \mathbf{z}) \exp\left(\frac{1}{\beta} r(\mathbf{x}, \mathbf{z})\right)$. Then, we can obtain:

$$
\begin{aligned}
&\min_\theta \mathbb{E}_{\mathbf{z} \sim \mathcal{N}(0,\mathbf{I})} \mathbb{E}_{p_{\text{fake}}^\theta} \left[ \log \frac{p_{\text{fake}}^\theta(\mathbf{x} \mid \mathbf{z})}{\frac{1}{Z(\mathbf{z})} p_{\text{fake}}^{\theta_0}(\mathbf{x} \mid \mathbf{z}) \exp\left(\frac{1}{\beta} r(\mathbf{x}, \mathbf{z})\right)} - \log Z(\mathbf{z}) \right] \\
&= \min_\theta \mathbb{E}_{\mathbf{z} \sim \mathcal{N}(0,\mathbf{I})} \mathbb{E}_{p_{\text{fake}}^\theta} \left[ \log \frac{p_{\text{fake}}^\theta(\mathbf{x} \mid \mathbf{z})}{p_{\text{fake}}^{\theta^*}(\mathbf{x} \mid \mathbf{z})} - \log Z(\mathbf{z}) \right], \\
&= \min_\theta \mathbb{E}_{\mathbf{z} \sim \mathcal{N}(0,\mathbf{I})} \left[ \mathbb{D}_{\text{KL}} \left( p_{\text{fake}}^\theta(\mathbf{x} \mid \mathbf{z}) \| p_{\text{fake}}^{\theta^*}(\mathbf{x} \mid \mathbf{z}) \right) - \log Z(\mathbf{z}) \right]
\end{aligned}
\tag{17}
$$

where $p_{\text{fake}}^{\theta^*}(\mathbf{x} \mid \mathbf{z}) = \frac{1}{Z(\mathbf{z})} p_{\text{fake}}^{\theta_0}(\mathbf{x} \mid \mathbf{z}) \exp\left(\frac{1}{\beta} r(\mathbf{x}, \mathbf{z})\right)$. We find that the previous equation is to optimize the KL divergence and obtain the optimal parameter as follows:

$$p_{\text{fake}}^\theta(\mathbf{x} \mid \mathbf{z}) = p_{\text{fake}}^{\theta^*}(\mathbf{x} \mid \mathbf{z}) = \frac{1}{Z(\mathbf{z})} p_{\text{fake}}^{\theta_0}(\mathbf{x} \mid \mathbf{z}) \exp\left(\frac{1}{\beta} r(\mathbf{x}, \mathbf{z})\right) \tag{18}$$

### A.2  Direct Alignment Algorithm

We follow the represenative direct alignment algorithm Typically, fine-tuning Large Language Models (LLMs) via reinforcement learning first requires training a reward model using human-preferred data pairs, denoted as $(\mathbf{x}_w, \mathbf{x}_l)$, representing human-preferred and human-dispreferred images in our setting. The reward model can be trained on image pairs with Bradley-Terry (BT) models [3]:

$$p(\mathbf{x}_w \succ \mathbf{x}_l \mid \mathbf{z}) = \frac{\exp(r(\mathbf{x}_w, \mathbf{z}))}{\exp(r(\mathbf{x}_w, \mathbf{z})) + \exp(r(\mathbf{x}_l, \mathbf{z}))}. \tag{19}$$

We can obtain the reward model based on the optimal policy as follows:

$$r(\mathbf{x}, \mathbf{z}) = \beta(\log \frac{p_{\text{fake}}^{\theta^*}(\mathbf{x}, \mathbf{z})}{p_{\text{fake}}^{\theta_0}(\mathbf{x}, \mathbf{z})} - \log Z(\mathbf{z})). \tag{20}$$

Then, we can get the BT model with the format of optimal policy:

$$p^*(\mathbf{x}_w \succ \mathbf{x}_l \mid \mathbf{z}) = \frac{1}{1 + \exp\left(\beta \log \frac{p_{\text{fake}}^{\theta^*}(\mathbf{x}_l \mid \mathbf{z})}{p_{\text{fake}}^{\theta_0}(\mathbf{x}_l \mid \mathbf{z})} - \beta \log \frac{p_{\text{fake}}^{\theta^*}(\mathbf{x}_w \mid \mathbf{z})}{p_{\text{fake}}^{\theta_0}(\mathbf{x}_w \mid \mathbf{z})}\right)} \tag{21}$$

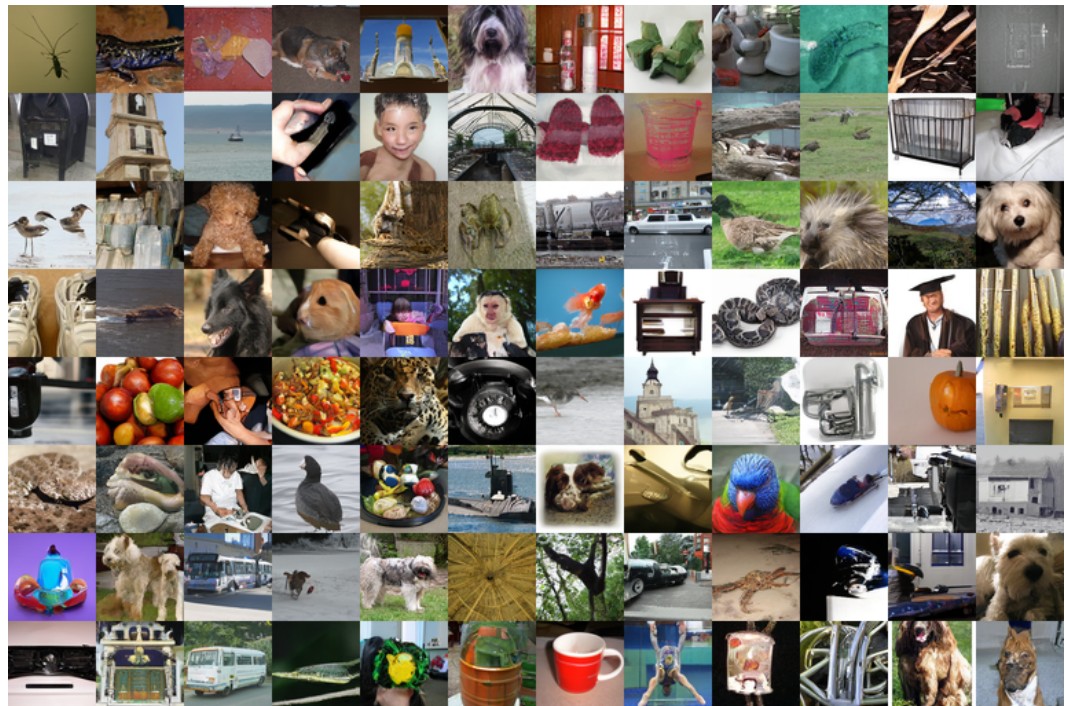

Figure 5: Images generated by one-step diffusion models trained with CED on ImageNet.

To avoid the complexity of two-step training, [41] introduces a direct optimization approach, replacing the optimal parameter $\theta^*$ with trainable parameters $\theta$. This strategy integrates the traditional two-stage process—first training reward models and subsequently training reinforcement learning models using these reward models—into a unified, single-step training pipeline:

$$\min_\theta -\log p\left(\mathbf{x}_w \succ \mathbf{x}_l \mid \mathbf{z}\right) = \min_\theta -\log \frac{1}{1 + \exp\left(\beta \log \frac{p_{\text{fake}}^{\theta^*}(\mathbf{x}_l|\mathbf{z})}{p_{\text{fake}}^{\theta_0}(\mathbf{x}_l|\mathbf{z})} - \beta \log \frac{p_{\text{fake}}^{\theta^*}(\mathbf{x}_w|\mathbf{z})}{p_{\text{fake}}^{\theta_0}(\mathbf{x}_w|\mathbf{z})}\right)}. \tag{22}$$

Therefore, our approach, viewed from the perspective of Density Ratio Estimation, shares conceptual similarities with direct alignment algorithms, as both leverage related techniques to unify the conventional two-step training process into a single step.

## B  Experiment Details

### B.1  Implementation Details

We present implementation and setup details of CED in this section. For experiments, we use the Adam optimizer with an effective batch size of 512 for CIFAR-10 and 1024 for ImageNet. Training is conducted on 2 NVIDIA A100 GPUs. We train at fixed square resolutions and use a learning rate 3e-5. Moreover, we perform 10000 steps warm-up training as demonstrated in Line 3 of Algorithm 1. Our code of CED is based on the implementation EDM [22].

## C  Additional Experiments

In this section, we present more results generated from CED on Cifar-10 and ImageNet as shown in Figure 5 and Figure 6, which further shows the effectiveness of our proposed method.

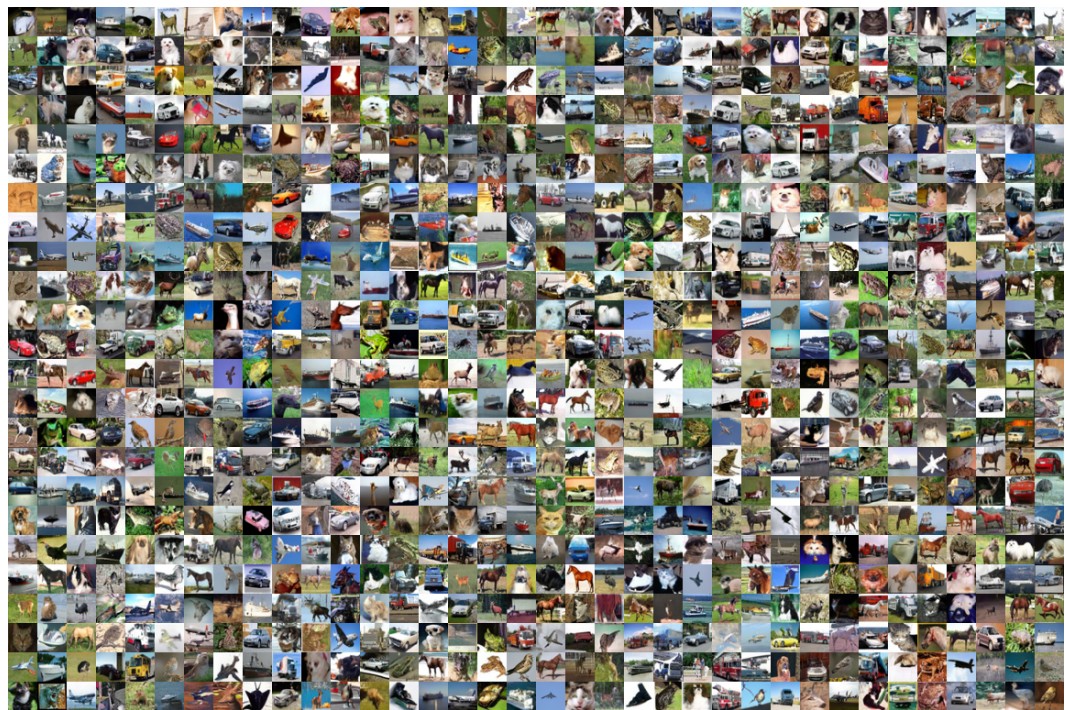

Figure 6: Images generated by one-step diffusion models trained with CED on Cifar-10.

## D    Limitations

This work introduces an algorithm for efficiently distilling knowledge from multi-step diffusion models into one-step diffusion models for image generation tasks. However, the current focus is limited to image generation, while other domains—such as scientific discovery and 3D reconstruction, which could also benefit from fast diffusion models to enhance the efficiency of their generative pipelines.

## E    Broader Impact

This work can be used to accelerate both image generation and text-to-image generation, enhancing the user experience in text-to-image systems. Furthermore, it can be extended to other domains for efficient high-quality sample generation, such as 3D reconstruction, audio synthesis, and scientific discovery.

