# OpenReview forum: "Simple Distillation for One-Step Diffusion Models"
_NeurIPS.cc/2025/Conference — NeurIPS 2025 poster_

### Official Review · Reviewer_x1Sm · 2025-06-21

**Clarity:** 3
**Significance:** 3
**Originality:** 3
**Rating:** 5
**Confidence:** 4

**Summary:**

This paper introduces Contrastive Energy Distillation (CED), a simple and efficient method to distill multi-step diffusion models into one-step generators. By leveraging a joint energy-based model and a contrastive loss inspired by Noise Contrastive Estimation, CED avoids auxiliary models and iterative training. It achieves competitive performance on CIFAR-10 and ImageNet.

**Questions:**

+ Please report on training time and compare it with other method like DMD, DMDv2, CD, SiD
+ Author should use same initial noise and same class input for sampling for Figure 4. So the reader can visually compare between methods

**Ethical Concerns:**

["NO or VERY MINOR ethics concerns only"]

**Final Justification:**

The author address my concerns well but I already give a high score with a huge faith in this work so I decide to keep inital score of **5**.

**Limitations:**

No limitations besides the weaknesses I mentioned above

**Paper Formatting Concerns:**

No issues

**Quality:**

3

**Strengths And Weaknesses:**

**Strengths**

+ CED eliminates the need for auxiliary score models and iterative updates, reducing memory usage and simplifying implementation.

+ On CIFAR-10 and ImageNet, CED achieves results close to or better than other one-step distillation methods, with faster convergence and lower resource demands.

+ The paper includes thorough ablation studies on energy functions and loss choices, offering insights into design decisions (e.g., LPIPS-Huber and hinge loss work well).

+ A quick glance in the provided code, I see it quite simple to implement so this is a plus.

**Weaknesses**

+ The method is only evaluated on 32×32 CIFAR-10 and 64×64 ImageNet; more difficult settings like text-to-image diffusion tasks are not explored.

+ If I understand correctly then the method relies the generator from previous iteration $G_{\theta'}$, which, while less burdensome than training the generator alongside with the auxiliary score model, still introduces an computational cost.  Therefore, I'm not convinced it is actually more efficient than DMD or previous approach as there is no training time comparison, also DMDv2 [1] has introduce some tricks to make it much more efficient.

+ Lack comparison with DMDv2, which I believe also report on same setting 32×32 CIFAR-10 and 64×64 ImageNet.

Reference:

[1] Yin et al. Improved Distribution Matching Distillation for Fast Image Synthesis. NeurIPS 2024

---

> ### Author Rebuttal · Authors · 2025-07-31
>
> We appreciate the reviewer's perception of our contributions to simplifying the training pipeline and being easy to implement, and we thank the reviewer for their insightful questions. Please find our detailed responses below:
>
> **Q1. The method is only evaluated on 32×32 CIFAR-10 and 64×64 ImageNet; more difficult settings like text-to-image diffusion tasks are not explored.**
>
> **A1.** Thanks for your suggestions! Due to limited time and computational resources, it is challenging for us to perform model distillation from scratch for text-to-image diffusion models initialized with multistep teacher diffusion models. Instead, we conduct fine-tuning experiments on SDXL-DMD2 on the text-to-image diffusion tasks. Specifically, we fine-tune SDXL-DMD2 using our proposed loss function on the Pick-a-Pic v2 dataset and evaluate its performance based on win rates across several reward models, including ImageReward, Aesthetic Score, and Pick Score. The win rate measures how often the fine-tuned model outperforms the original model based on different reward models, with a performance above 50% indicating an improvement. All evaluations are conducted on the test split of the Pick-a-Pic v2 dataset. Here are the following results corresponding to our current optimization steps:
>
> | Reward Model       | Pick Score | Aesthetic Score | ImageReward |
> |--------------|---------------|---------------------|---------------------|
> | CED         |   58.60  |           56.20          |  55.60  |
>
> This experiment demonstrates both the effectiveness and efficiency of our approach, as we successfully fine-tune the model using a single A100 GPU. Achieving a win rate above 50% indicates that our method consistently improves image quality in the complex scenario like text-to-image generation. Therefore, it shows the potential of our approach for more difficult settings like text-to-image diffusion tasks.
>
> **Q2. Therefore, I'm not convinced it is actually more efficient than DMD or previous approach as there is no training time comparison, also DMDv2 [1] has introduce some tricks to make it much more efficient.**
>
> **A2.** Thanks for your suggestions and comments! Our method relies on the distillation setting without real image data. However, DMDv2 operates under a different setting from ours, as it incorporates GANs and relies on real data to enhance performance during training. This makes a direct comparison with our method challenging. To fairly evaluate the efficiency of our approach, we instead compare it with another distillation method that, like ours, does not use real data—SiD. **The results are as follows: CED achieves an FID of 3.85 on CIFAR-10 using only 20 A100 GPU hours, whereas SiD requires 100 A100 GPU hours to reach an FID of 4.0.** These findings highlight the training efficiency of our method in achieving comparable performance with significantly lower computational cost.
>
> **Q3. Lack comparison with DMDv2, which I believe also report on same setting 32×32 CIFAR-10 and 64×64 ImageNet.**
>
> **A3.** Thanks for your suggestions and comments! DMDv2 operates under a fundamentally different setting from ours, as it incorporates GANs and leverages real data to boost performance during training. As a result, making a direct comparison with our method is not straightforward. However, we include comparisons with DMD—without the use of GAN techniques—in Tables 2 and 3. Our method demonstrates improved performance on CIFAR-10 and achieves comparable results on ImageNet, all while avoiding the need for additional models and iterative optimization. Moreover, we believe that integrating our method with GANs presents an interesting direction for further improvement. To explore this, we combined CED with a GAN-based approach similar to DMDv2 and evaluated it on CIFAR-10 using real image data. This resulted in an impressive FID of 1.85, demonstrating a significant performance boost over previous methods. However, due to time constraints, we were unable to conduct experiments on ImageNet, and we note that DMDv2 does not report results on CIFAR-10 for direct comparison. We plan to extend our experiments to ImageNet in future work. Nonetheless, these preliminary results highlight the strong potential of our method and its compatibility with state-of-the-art techniques to further enhance performance.
>
> **Q4. Please report on training time and compare it with other method like DMD, DMDv2, CD, SiD.**
>
> **A4.** Thanks for your suggestions! Since DMDv2 extends DMD by incorporating GANs to train one-step diffusion models with real data, it falls outside the scope of our setting, which focuses on distillation-based methods without access to real data. Moreover, the authors of DMD do not provide training code, making direct comparison difficult. Instead, we compare our method with SiD, which adopts a similar iterative training scheme, relies on additional models and claims faster convergence. Until now, we have the following results: CED achieves an FID of 3.85 on CIFAR-10 using only 20 A100 GPU hours, whereas SiD requires 100 A100 GPU hours to reach an FID of 4.0. Moreover, CD requires approximately 1000 A100 GPU hours on CIFAR-10 to achieve an FID of 3.55, whereas our proposed method reaches comparable performance with only about 30 A100 GPU hours. This significant reduction in computational cost further demonstrates the efficiency of our approach.
>
> **Q5. Author should use same initial noise and same class input for sampling for Figure 4. So the reader can visually compare between methods**
>
> **A5.** Thanks for your suggestions! Due to format constraints in the rebuttal, we will include these results in the revised version of the paper.

---

> > ### Comment · Reviewer_x1Sm · 2025-08-05
> >
> > The author address my concerns well but I already give a high score with a huge faith in this work so I decide to keep inital score of **5**.

---

> > > ### Author Response · Authors · 2025-08-05
> > >
> > > Thank you for recognizing our rebuttal—we sincerely appreciate your thoughtful feedback. Your positive feedback is really important to us!

---

### Official Review · Reviewer_SDf9 · 2025-07-02

**Clarity:** 3
**Significance:** 2
**Originality:** 3
**Rating:** 4
**Confidence:** 5

**Summary:**

This paper introduces Contrastive Energy Distillation (CED), a simple and efficient method to convert slow, multi-step diffusion models into fast, one-step image generators. The key idea is to frame the distillation as a contrastive learning task, where a student model learns to distinguish "positive" samples from a teacher model against "negative" samples from a previous version of itself. This approach eliminates the need for complex auxiliary models and iterative training, significantly reducing memory usage and training complexity. The authors provide theoretical justification for their method and show through experiments that CED achieves competitive image quality with far greater efficiency than comparable techniques.

**Questions:**

See Weaknesses.

**Ethical Concerns:**

["NO or VERY MINOR ethics concerns only"]

**Final Justification:**

The rebuttal by authors addresses some of my concerns, thus I raise the score to 4, assuming that the author completely eliminates the overclaiming/misleading writing in the paper, especially in Figure 1 and Tables 2-3.

**Limitations:**

See Weaknesses.

**Paper Formatting Concerns:**

No formatting issues in this paper.

**Quality:**

3

**Strengths And Weaknesses:**

Strengths:
- The core idea of using a contrastive Energy objective to traina  one-step generator is interesting.
- The paper provides a compelling theoretical connection between the proposed CED objective and KL divergence.
- Experiments show competitive performance.

Weaknesses:
- This paper seems to overclaim multiple points. In multiple places, the paper claims CED is a distillation algorithm with the property of Student Models Only. However, it still requires the teacher to generate noise-image pairs, which is expensive and not student-only. Besides, CED does not only use 1/3 or 1/2 memory compared to score distillation and consistency distillation. The memory cost of distillation includes model weights, optimizer state, and so on.
-  The performance gap between SiD and CED is still notably large.
- Lack an important variant: remove the negative phase in the CED objective, which is closely related to InstaFlow [1].
- Lack of comparison to some recent SOTA with one trainable model, e.g., IMM [2] and shortcut model [3].

Minor:
- The title of Section 4.3 would be clearer and better if it were changed to 'Relation to Reverse KL Divergence'. Specifically, the usage of reverse KL in diffusion distillation is first proposed in VSD and Diff-Instruct. It is somewhat inappropriate to use DMD to represent reverse KL divergence.

[1] InstaFlow: One Step is Enough for High-Quality Diffusion-Based Text-to-Image Generation, ICLR 2023.

[2] Inductive Moment Matching, ICML 2025.

[3] One Step Diffusion via Shortcut Models, ICLR 2025.

---

> ### Author Rebuttal · Authors · 2025-07-31
>
> We sincerely thank the reviewer for the comments and suggestions. Please see our clarifications below:
>
> **Q1. In multiple places, the paper claims CED is a distillation algorithm with the property of Student Models Only. However, it still requires the teacher to generate noise-image pairs, which is expensive and not student-only.**
>
> **A1.** Thanks for your comments and suggestions! We would like to kindly remind the reviewer that we have demonstrated that **“Note that these categorizations are based specifically on the models involved during the training of student generators.” in line 283.** We do not claim that our method avoids using teacher models entirely; rather, we emphasize that the teacher model is not required during the training process. We will further clarify this in the revised paper.
>
> Moreover, the time cost for pre-sampling before training is significantly lower than the overall training cost. **For example, CED achieves an FID of 3.85 on CIFAR-10 using only 20 A100 GPU hours, while SiD requires 100 A100 GPU hours to reach an FID of 4.0. In our case, generating the pre-sampled data for CIFAR-10 takes only about 2–3 A100 GPU hours, which is negligible compared to the large training time gap between CED and SiD.** The key advantage of pre-sampling is that it eliminates the need to generate samples during each training step. Instead, we can reuse the pre-sampled data throughout training, which significantly improves overall efficiency.
>
> **Q2. Besides, CED does not only use 1/3 or 1/2 memory compared to score distillation and consistency distillation. The memory cost of distillation includes model weights, optimizer state, and so on.**
>
> **A2.** Thanks for your comments and suggestions! We think that the memory savings—specifically the reduction of 1/3 to 1/2 in model loading memory—are mentioned in Section 5.2. For example, line 304 states: “significantly reducing memory consumption of loading models by approximately 50% during training,” and line 309 notes: “thus achieving a 66% reduction in memory usage of loading models.” These statements highlight our focus on reducing memory usage related to model loading. We will further clarify this in the revised version. Moreover, we acknowledge that the description in Figure 1 may have caused some confusion. We apologize for this and will clarify the figure and explicitly emphasize the 1/3 to 1/2 memory savings in model loading more clearly in the revised version. Additionally, methods like DMD and SiD require iterative training across multiple models, which can lead to increased memory usage due to the need to store optimizer states. In this paper, we focus on the practical costs associated with model loading and the complexity of training pipelines, which often introduce more hyperparameters and result in slower convergence rates. **Moreover, we believe the reviewer should not overlook our contribution to improving the training efficiency of current baselines. Unlike consistency-based distillation techniques, our method does not require access to the teacher model's outputs.** Additionally, **we avoid the need to optimize auxiliary fake score models**, which would otherwise require extra memory and additional optimization steps. We propose a new perspective for designing a simple and efficient distillation technique for one-step generative models distilled from multi-step diffusion models—without relying on a complex training framework.
>
>
> **Q3. The performance gap between SiD and CED is still notably large.**
>
> **A3.** Thanks for your comments and suggestions! We would like to clarify that our primary contribution is not to propose a state-of-the-art method for distilling multi-step diffusion models into one-step generators without real data. Rather, our goal is to introduce a new perspective that challenges the existing complex training frameworks with extra models and iterative optimization. We believe our approach offers a simpler alternative that significantly reduces computational cost and training time, while providing fresh insights into the design of efficient distillation techniques. Moreover, as mentioned in our previous response, our method demonstrates significantly lower computational cost—for example, **CED achieves an FID of 3.85 on CIFAR-10 using only 20 A100 GPU hours, whereas SiD requires 100 A100 GPU hours to reach an FID of 4.0.** This highlights the efficiency of our approach and reinforces its potential as a promising alternative distillation training pipeline.
>
>
> **Q4. Lack an important variant: remove the negative phase in the CED objective, which is closely related to InstaFlow [1].**
>
> **A4.** Thanks for your suggestions. We have removed the negative phase in the CED objective and get the following results:
>
> |    Cifar10   | FID |
> |--------------|---------------|
> | w/o Negative         |   3.99  |
> |      CED    |      **2.97**         |
>
> We observe that our method significantly outperforms the variant without the negative gradient component, further demonstrating the effectiveness of our approach.
>
> **Q5. Lack of comparison to some recent SOTA with one trainable model, e.g., IMM [2] and shortcut model [3].**
>
> **A5.** Thanks for your suggestions! We would first like to clarify that our method is designed for distillation settings that do not rely on real image data, whereas IMM and shortcut models are trained using real image data. Additionally, since the shortcut models do not report results on CIFAR-10 or ImageNet, we include the results of IMM on CIFAR-10 as reported in their paper:
>
> |    Cifar10   | FID |
> |--------------|---------------|
> | IMM         |   3.20  |
> |      CED-DRE    |      **2.97**         |
>
> We observe that our method outperforms IMM, even though IMM is trained on real image data.
>
> **Q6. The title of Section 4.3 would be clearer and better if it were changed to 'Relation to Reverse KL Divergence'.**
>
> **A6.** Thanks for your suggestions! We will modify this in the revised version.

---

> ### Author Response · Authors · 2025-08-04
> **Dear NeurIPS Reviewer SDf9: we understand that you maybe busy, so we would greatly appreciate it if you could check out our rebuttal.**
>
> Dear NeurIPS Reviewer SDf9:
>
> We gratefully appreciate your time in reviewing our paper and your insightful comments. We made our greatest efforts to address your concerns in the rebuttal. The reviewer's comments primarily involved requests for clarification and appeared to reflect some misunderstandings, which we have addressed point by point in our responses. As the discussion period draws to a close, we would like to check if there are any remaining questions or concerns we can clarify. If our responses have resolved your concerns, we would greatly appreciate your consideration in updating your score. Thank you very much once again; we are extremely grateful.
>
> Best regards
>
> The authors of "Simple Distillation for One-Step Diffusion Models"

---

> > ### Author Response · Authors · 2025-08-08
> > **Kind reminder to reviewer SDf9: the author-reviewer discussion period is coming to an end.**
> >
> > Dear Reviewer SDf9,
> >
> > We want to sincerely thank you again for your time and comments!
> >
> > We have made extensive efforts to address your comments and carefully considered your suggestions point-by-point and added results for the efficiency of CED, ablation studies you mention and comparing with IMM in the rebuttal. Moreover, we would like to reiterate our main contributions to address your concerns. We believe our method is important for simplifying the training pipeline by distilling multi-step diffusion models into a one-step diffusion approach. As noted by Reviewer x1Sm, “A quick glance at the provided code shows it is quite simple to implement, so this is a plus." We hope you will reconsider our paper in light of our simple and efficient algorithm, especially when compared to previous methods that require iterative optimization and the loading of multiple components.
> >
> > We would like to kindly remind you that we are approaching the end of the author-reviewer discussion. In light of our rebuttal, we kindly ask if you could consider increasing your score. Thank you very much once again, and we look forward to hearing back from you if you have further comments.
> >
> > Thank you very much for your time.
> >
> > Best regards,
> >
> > The Authors of "Simple Distillation for One-Step Diffusion Models"

---

> > > ### Comment · Reviewer_SDf9 · 2025-08-09
> > >
> > > Thanks for the detailed reply. After carefully reading the response and other reviews, I still have some concerns:
> > >
> > > 1) Although there is an explanation in Line 283, the table is still misleading. I still feel the overclaim issues exist in the current version of this paper.
> > > 2) Regarding efficiency. I know generating samples on CIFAR-10 is efficient. However, it is highly expensive in text-to-image tasks. For example, DMD2 mentioned that the construction of 12M $\epsilon, x$ on the SDXL backbone requires 700 A100 days.
> > > 3) Regarding memory. The memory and time cost of model loading is less meaningful. In practice, people care about the overall training memory cost.
> > >
> > > Given the above concerns, I tend to keep my original rating. But I would lower my confidence and would decide the final rating in the internal discussion among reviewers.

---

> ### Author Response · Authors · 2025-08-09
>
> Thank you for your insightful questions and feedback!
>
> **Q1. Although there is an explanation in Line 283, the table is still misleading. I still feel the overclaim issues exist in the current version of this paper.**
>
> **A1.** Thanks for your suggestions! We have already included it in Line 283 to avoid confusion. We will also incorporate these points into the table in the revised version to avoid any potential confusion. We believe this does not affect our main contribution of introducing a new and simple training pipeline, and we have also demonstrated its faster convergence in A2 of x1Sm.
>
> **Q2. Regarding efficiency.**
>
> **A2.** Thanks for your coments! We agree with your concern that generating images with large models can be cumbersome. However, our method can be further improved without requiring the generation of too many images, as DMD2 is essentially an advanced version of DMD. **To clarify, DMD2 incorporates GAN techniques during training but differs from our approach in key aspects—for instance, we do not use real images during training.** From our observations, GAN helps DMD2 converge faster than its predecessor, DMD, likely due to leveraging the information from real images. Therefore, we believe that DMD, when trained under the same settings as our algorithm, would require significantly more time to converge. **As discussed in our response to Reviewer x1Sm, we also integrate GAN techniques into our method, achieving an FID of 1.85 with only ~30 A100 GPU hours.** For this training setup, we first optimize our objective on both noisy and generated images, followed by training the one-step generator with GANs. **This allows us to sample partial images and integrate GAN-based techniques, such as DMD2, to leverage information from real images.** For instance, on the LAION dataset, we can randomly sample 10% of the prompts for full training (objective + GAN) and, for the remaining prompts, apply only the GAN loss to train the one-step generator. **Intuitively, compared with directly using GAN, our method stabilizes generator training by not relying solely on gradients propagated through the discriminator. Incorporating our loss also implicitly leverages the DMD loss, as discussed in Section 4.3. Therefore, this method is a potential approach to reduce the sampling time.**
>
> Due to the limited time for rebuttal and constraints on computational resources, we were unable to implement this in text-to-image settings. However, we have already achieved promising results when combining our method with GANs. If time permits, we plan to conduct a quick experiment on CIFAR-10, where we sample fewer images, optimize the GAN objective for more steps, and then apply our objective to the CIFAR-10 dataset to verify the idea of reducing the number of sampling images.
>
> Moreover, compared with DMD2, if DMD2 is already trained with a one-step generator, it still requires storing additional fake score models for fine-tuning. This will lead to more computational cost and demands extensive hyperparameter search when optimizing two components for fine-tuning. In contrast, our method enables straightforward fine-tuning of text-to-image models to improve quality, as demonstrated in A3 of Reviewer deet. Specfically, when following the hyperparameters used in DMD2, its complex structure makes performance gains difficult to achieve while our method can be easily used to improve the quality of images. **Therefore, we believe our training pipeline retains a significant advantage for future fine-tuning compared with DMD2.**
>
> Therefore, we consider the text-to-image setting as future work, where we can explore different strategies to address sampling efficiency issues for text-to-image settings, which are somewhat beyond our current scope. We believe our method serves as a strong starting point for avoiding complex structures and can be further applied and refined in more complex settings.
>
> **Q3. Regarding memory.**
>
> **A3.** Thanks for your comments! As you noted in the previous question, memory usage includes optimizer states, model weights, and potentially activation memory. Compared with DMD, our approach saves approximately one-third of the model weights, half of the optimizer states, and half of the activation memory, since DMD requires loading three models and optimizing two of them iteratively. We respectfully disagree with the characterization of these memory savings as “meaningless,” as they can reduce training time and improve efficiency as shown in A2 of x1Sm. Our simple training pipeline achieves these benefits to converge faster compared with previous distillation methods such as DMD and SiD.
>
> Hope these responses can resolve your concern!

---

> > ### Comment · Reviewer_SDf9 · 2025-08-09
> >
> > Thanks for further clarification. I would be willing to raise the score to 4, assuming that the author **completely eliminates the overclaiming/misleading writing in the paper**, especially in Figure 1 and Tables 2-3.

---

> ### Author Response · Authors · 2025-08-09
>
> Thanks for your comments and raising the score! We greatly appreciate your positive feedback and will revise the writing to resolve the confusion in Figure 1 and Tables 2-3 in the paper.

---

### Official Review · Reviewer_e2GZ · 2025-07-03

**Clarity:** 2
**Significance:** 3
**Originality:** 3
**Rating:** 4
**Confidence:** 4

**Summary:**

This paper proposes Contrastive Energy Distillation (CED), a simple method for distilling one-step diffusion models. CED removes the need for introducing fake and real score models during the iterative training process by introducing a joint energy-based model that connects the one-step generator and fake score function. The method achieves competitive performance with improved memory efficiency compared to existing distillation approaches.

**Questions:**

1.	What is the distillation efficiency (e.g., convergence speed) compared to methods that use a fake score model? Is there any direct comparison?
	2.	Since the CED training method requires pre-generated data, it would be helpful to see how different methods also requires pre-generated data.
	3.	To better understand the core idea of the paper: Is it correct to say that previous methods rely on both a fake score model and a real score model to learn an energy metric, whereas CED introduces a joint energy-based framework that explicitly models the energy, allowing the fake and real score models to be removed? Please correct me if I’m misunderstanding.

**Ethical Concerns:**

["NO or VERY MINOR ethics concerns only"]

**Final Justification:**

The authors have addressed most of my concerns, and I am inclined to maintain my current score.

**Limitations:**

Discussed in Weakness and Questions.

**Quality:**

3

**Strengths And Weaknesses:**

Strengths:
The paper addresses a practical problem by simplifying the distillation process for diffusion models. The proposed method is simple yet effective, achieving satisfying performance while removing the need for both the fake score model and the real score model during training, which improves memory efficiency. It is also interesting to see that the distillation remains effective despite removing these components.

Weakness:
1. The paper’s structure, particularly Sections 4.1 to 4.3, is not very reader-friendly and needs significant improvement in writing clarity. For example, the motivation and intuition for introducing the joint energy-based function in Section 4.1 should be explained more clearly.
2. Additionally, the connection and distinction between DMD and CED remain unclear despite the extensive discussion in Section 4.3.
3.  It would also be helpful to include more results on distillation efficiency (convergence speed) compared to other methods, as the current paper only provides a CD comparison in Figure 2.

---

> ### Author Rebuttal · Authors · 2025-07-31
>
> We sincerely thank the reviewer for the comments and suggestions. Please see our clarifications below:
>
> **Q1. The paper’s structure, particularly Sections 4.1 to 4.3, is not very reader-friendly and needs significant improvement in writing clarity. For example, the motivation and intuition for introducing the joint energy-based function in Section 4.1 should be explained more clearly.**
>
> **A1.** Thanks for your suggestions! Our motivation stems from the training process used in previous distillation-based methods, which typically involve an additional model to first train a fake score model $p_{\text{fake}, \phi}$ that approximates the score function of samples generated by the one-step generator $G_\theta$.  However, the ultimate goal is to match the distribution of samples from the teacher model. To capture the relationships among these three distributions—those of the one-step generator, the fake score model, and the teacher model—we are inspired to formulate the process as a joint energy-based model.
> Specifically, we calibrate the fake score distribution by modeling the discrepancy between the samples from the one-step generator (which we treat as the energy in our framework) and the samples from the learned fake score model in Equation (4). This is because the learned fake score model cannot accurately capture the target distribution. Therefore, we use the energy function to calibrate the original fake score distribution, enabling it to better align with the distribution of samples generated by the one-step generator $G_\theta$. Since our final objective is to align with the teacher model, we construct this joint formulation that equals the distribution of teacher models and use NCE to optimize this energy-based model to achieve this goal. In the revised version, we will further clarify our high-level motivation and intuition about modeling the relationships among the three models. And we aim to simplify the training process by eliminating the need for intermediate fake score models, highlighting the core idea of our approach more clearly.
>
> **Q2. Additionally, the connection and distinction between DMD and CED remain unclear despite the extensive discussion in Section 4.3.**
>
> **A2.** Thanks for your suggestions! Before addressing this question, we would like to first clarify that the original DMD approach **requires additional model training and iterative optimization**. In contrast, our proposed CED method **does not involve loading extra models and avoids iterative training altogether**, making it fundamentally different from the original DMD. **Furthermore, we derive a new objective from the persepctive of DMD, as shown in Equation (13), which simplifies the original loss and removes the need for complex training procedures.** Notably, we observe that this objective implicitly optimizes DMD and shares a similar format with CED when a specific form of the energy function is defined. The difference between CED and CED-DRE in Table 2 is based on different formats of energy functions. Specifically, CED is based on $\boldsymbol{E}\left(\mathbf{x}, G_\theta(\mathbf{z})\right)=\left\|x-G_\theta(\mathbf{z})\right|^{2}$ and CED-DRE is baed on $\boldsymbol{E}\left(\mathbf{x}, G_\theta(\mathbf{z})\right)=-\left(\left\|\mathbf{x}-G_\theta(\mathbf{z})\right\|^{2}-\left\|\mathbf{x}-G_{\theta^{\prime}}(\mathbf{z})\right\|^{2}\right)$ with L2 distance.
>
> **Q3. It would also be helpful to include more results on distillation efficiency (convergence speed) compared to other methods, as the current paper only provides a CD comparison in Figure 2. What is the distillation efficiency (e.g., convergence speed) compared to methods that use a fake score model? Is there any direct comparison?**
>
> **A3.** Thanks for your questions! We conducted an experiment to compare the convergence speed of our method with SiD, a strong baseline that requires training fake score models. **Our results show that CED achieves an FID of 3.85 on CIFAR-10 using only 20 A100 GPU hours, whereas SiD requires 100 A100 GPU hours to reach an FID of 4.0.** This demonstrates that our method converges significantly faster than methods like SiD that rely on training auxiliary score models.
>
>
> **Q4. Since the CED training method requires pre-generated data, it would be helpful to see how different methods also requires pre-generated data.**
>
> **A4.** Thanks for your questions! We use pre-generated data to avoid the need for loading additional models during training and to enable efficient reuse of data without generating new samples at each optimization step. In contrast, consistency model-based distillation techniques require access to the outputs of teacher models, making it necessary to load those models and difficult to rely on pre-generated data. Similarly, methods that involve training auxiliary fake score models also depend on teacher model outputs and require additional model training, which further prevents the use of pre-generated data.
> In our case, the key advantage lies in the design of our loss function, which does not require the output from teacher models—only samples generated by them. This unique characteristic allows us to effectively use pre-generated data, significantly reducing memory consumption and simplifying the training pipeline.
>
> **Q5. To better understand the core idea of the paper: Is it correct to say that previous methods rely on both a fake score model and a real score model to learn an energy metric, whereas CED introduces a joint energy-based framework that explicitly models the energy, allowing the fake and real score models to be removed? Please correct me if I’m misunderstanding.**
>
> **A5.** Thanks for your question! Previous methods model the fake score model, real score model, and one-step generator as separate components, requiring them to optimize each part individually during DMD training. **In contrast, our approach captures the relationships among these components within a single joint energy-based model and eliminates the need for explicitly modeling or training fake score models.**
> Our connection to prior methods, such as DMD, is established through the analysis presented in Section 4.3. We demonstrate that optimizing our CED objective is equivalent to implicitly optimizing DMD under a specific formulation of the energy function. This unified perspective not only simplifies the training process but also provides a deeper theoretical understanding of how our method relates to and improves upon previous approaches.

---

> > ### Author Response · Authors · 2025-08-04
> > **Dear NeurIPS Reviewer e2GZ: we understand that you maybe busy, so we would greatly appreciate it if you could check out our rebuttal.**
> >
> > Dear Neurips Reviewer e2GZ,
> >
> > We sincerely appreciate your time and the thoughtful feedback you provided in reviewing our paper. We have made great efforts to address your comments and believe our responses adequately resolve the concerns raised. The points mentioned primarily relate to clarification and do not undermine the core contributions of our work. We believe that these issues can be effectively addressed in the final version, and we are grateful for your constructive suggestions.
> >
> > We would like to confirm whether there are any other clarifications they would like for the discussion period nears its end. If the reviewer's concerns are clarified, we would be grateful if the reviewer could update the score. Many thanks for your time; we are extremely grateful.
> >
> > The authors of “Simple Distillation for One-Step Diffusion Models”

---

### Official Review · Reviewer_deet · 2025-07-05

**Clarity:** 3
**Significance:** 3
**Originality:** 2
**Rating:** 3
**Confidence:** 3

**Summary:**

The paper introduces Contrastive Energy Distillation (CED), which uses a joint energy-based model and Noise Contrastive Estimation to distill multi-step diffusion models into efficient one-step generators. CED eliminates auxiliary models and iterative training, reducing memory usage by 1/2 while achieving competitive image quality and requiring fewer training steps than existing approaches.

**Questions:**

1. How does CED perform against recent non-distillation approaches? The authors are suggested to include direct comparisons with at least 2-3 non-distillation one-step methods on the same datasets. Clearly articulate when distillation is preferable.
2. How does CED scale to high-resolution images (e.g., 512×512, 1024×1024) and complex datasets beyond CIFAR-10/ImageNet 64×64? Are the memory advantages maintained at scale? The authors should provide experiments on at least one high-resolution dataset (ImageNet 256×256 or 512×512) with memory usage analysis.
3. The connection to distribution matching distillation (Section 4.3) is valuable, but is the theoretical contribution sufficiently novel beyond applying existing NCE frameworks to this setting? The authors should clarify the novel theoretical challenges specific to diffusion distillation that necessitate the proposed energy-based formulation.

**Ethical Concerns:**

["NO or VERY MINOR ethics concerns only"]

**Final Justification:**

I recognize the contributions of this paper, but I still feel that the restricted evaluation somewhat limits the paper's quality, particularly in its lack of comprehensive evaluation on larger image resolutions and a comprehensive comparison with non-distillation approaches. Given these considerations, I raised my score, but still with some reservations.

**Limitations:**

Yes

**Quality:**

2

**Strengths And Weaknesses:**

Strengths：
1. Clear presentation: Well-structured with clear motivation, problem statement, and generally clear mathematical exposition.
2. Theoretical foundation: Provides a principled connection between CED and distribution matching distillation, offering theoretical justification for the approach.
3. Broad applicability: The energy-based formulation could potentially extend to other generative modeling domains beyond images.

Weaknesses：
1. Incremental contribution: Primarily combines existing techniques (EBMs, NCE), limiting theoretical novelty.
2. While the paper provides thorough comparisons within distillation methods, the evaluation scope is unnecessarily narrow. Recent work has demonstrated that non-distillation approaches can achieve competitive one-step generation quality, such as shortcut models [Frans et al., 2024]. The paper lacks a clear articulation of distillation's specific scenario constraints (no access to original training data, goal of accelerating a specific pre-trained teacher), and fails to provide broader context by comparing against these alternative paradigms in evaluation.
3. Dataset scope: Evaluations are limited to relatively simple datasets (CIFAR-10, ImageNet 64x64); unclear how it scales to high-resolution or complex domains.
4. Modest performance gains: While memory-efficient, quality improvements are incremental compared to the computational savings achieved.

[Frans et al. 2024] "One Step Diffusion via Shortcut Models" ICLR 2025.

---

> ### Author Rebuttal · Authors · 2025-07-31
>
> We gratefully appreciate your time in reviewing our paper. We would like to clarify some misunderstandings regarding our approach.
>
> **Q1. Incremental contribution: Primarily combines existing techniques (EBMs, NCE), limiting theoretical novelty.**
>
> **A1.** Thanks for your comments! We believe that the Reviewer has important misunderstandings of our method. **Our method is not merely a straightforward combination of EBMs and NCE** and the core contribution and novelty lie in simplifying the training pipeline for distilling multi-step diffusion models into a single-step generator. Specifically, previous methods derive their loss from distribution matching [5] or score identity distillation [6]. However, these methods can’t avoid computing extra (or "fake") score models and optimizing the distillation process through iterative updates, which needs to load multiple models and significantly increase training complexity and resource demands. Our method simplifies the distillation process by rethinking it through the perspective of a novel joint energy-based model to connect fake score models, teacher models and one-step generators. By leveraging NCE alongside an approximation of the original loss, we derive an objective that avoids the need for additional models or iterative optimization. **We believe that our formulation of a joint energy-based model, combined with the adaptation of NCE to eliminate reliance on fake score models, is a novel contribution for simplifying the distillation of one-step generators.**
>
> Moreover, **we offer a novel and distinct perspective for understanding the loss as discussed in Section 4.3.** Notably, the derivation in Section 4.3 does not rely on any prior assumptions about the joint energy-based model. Instead, we derive a similar loss from the perspective of distribution matching-based distillation. Interestingly, the loss obtained from this distribution matching view shares a similar form with the one derived using our proposed joint EBM and NCE framework. **This unified perspective not only provides novel theoretical insight but also helps explain the effectiveness of our method by establishing a clear connection to existing distribution matching-based distillation approaches.**
>
> In summary, our method is not a simple combination of EBMs and NCE. **The core novelty lies in our analysis, which reformulates distillation-based methods as a joint energy-based model.** This perspective allows us to simplify the entire training pipeline and establish a clear connection between our proposed algorithm and distribution matching losses—commonly used in distillation—providing a deeper understanding of why our method is effective. As a result, our approach introduces a novel and simple training framework that offers new insights for designing distillation algorithms, enabling efficient one-step generators distilled from multi-step teacher diffusion models, while avoiding the complexity of prior methods that require iterative procedures and multiple components.
>
> **Q2. The evaluation scope is unnecessarily narrow. The paper lacks a clear articulation of distillation's specific scenario constraints.**
>
> **A2.** Thanks for your comments and suggestions! Our method is based on a distillation technique that does not rely on high-quality real data, which is a key focus of this paper. We aim to simplify the distillation pipeline specifically in settings where real data is not available. We will further clarify this point in the revised version, particularly in the experimental setup. We believe our method addresses a different scenario compared to works like [3], which focus on training with real data. To further verify the effectiveness of our approach, we also compare our method with some baselines trained with real data in A5.
>
>
> **Q3. Dataset scope: Evaluations are limited to relatively simple datasets (CIFAR-10, ImageNet 64x64); unclear how it scales to high-resolution or complex domains.**
>
> **A3.** Thanks for your suggestions! Due to the limit of time, it’s hard to realize full dataset training on these large datasets with our limited computational resources. To verify the effectiveness of our approach on complex tasks, **we conduct an experiment on the fine-tuning one step generator SDXL-DMD2 [4] to generate high-resolution images at 1024×1024 resolution** and following the evaluation method in Diffusion-DPO [2]. Specifically, we fine-tune SDXL-DMD2 using our proposed loss function on the Pick-a-Pic v2 dataset and evaluate its performance based on win rates across several reward models, including ImageReward, Aesthetic Score, and Pick Score. The win rate measures how often the fine-tuned model outperforms the original model based on different reward models, with a performance above 50% indicating an improvement. All evaluations are conducted on the test split of the Pick-a-Pic v2 dataset. Here are the following results **corresponding to our current optimization steps (the training is not converged)**:
>
> | Reward Model       | Pick Score | Aesthetic Score | ImageReward |
> |--------------|---------------|---------------------|---------------------|
> | CED         |   58.60  |           56.20          |  55.60  |
>
> This experiment demonstrates both the effectiveness and efficiency of our approach, as we successfully fine-tune the model using a single A100 GPU. Achieving a win rate above 50% indicates that our method consistently improves image quality in the complex scenario like text-to-image generation. In contrast, the DMD2 [4] framework—which focuses on training text-to-image diffusion models for one-step generation using real data—requires significantly more resources, including 8 compute nodes and multiple model downloads, as shown in their Github Repo. We also attempted to fine-tune SDXL-DMD2 on a single node with their loss and found the training process to be extremely slow. **Given the current optimization steps of DMD2 loss, we observed a PickScore win rate of only 52.80%, with aesthetic and image reward scores falling below 50%.** These results indicate that the DMD2 framework is challenging to train during the fine-tuning stage, particularly due to the difficulty of adjusting hyperparameters across multiple components. These results highlight the efficiency and potential of our method, particularly for complex scenarios such as text-to-image generation.
>
> **Q4. Modest performance gains: While memory-efficient, quality improvements are incremental compared to the computational savings achieved.**
>
> **A4.** Thanks for your comments! Our method outperforms previous distillation approaches that require loading teacher diffusion models during training and converges faster than methods such as consistency model distillation, as discussed in Section 5.2 and illustrated in Figure 2. This demonstrates both the effectiveness and efficiency of our approach.
>
> **Q5. How does CED perform against recent non-distillation approaches? The authors are suggested to include direct comparisons with at least 2-3 non-distillation one-step methods on the same datasets. Clearly articulate when distillation is preferable.**
>
> **A5.** Thanks for your suggestions! To address when distillation is preferable: if high-quality image data is not available, current distillation techniques are particularly suitable, as they are typically performed in settings that do not require real image data. Moreover, we compare our method with the latest approach, IMM, accepted at ICML 2025. It is important to note that this method is trained on real image data, unlike our setting, which does not rely on real data and here are corresponding results (results are from IMM) for one-step generation:
>
> |    Cifar10   | FID |
> |--------------|---------------|
> | ICT  |   2.83      |
> | ECT         |    3.60  |
> | sCT         |    2.97  |
> | IMM         |   3.20  |
> |      CED-DRE    |      2.97         |
>
> As shown in the table, our model achieves the second-best performance and has results comparable to the best-performing method. **Note that our method doesn’t rely on real data or loading additional teacher models during training.** This further demonstrates the effectiveness and potential of our approach for distilling one-step generators from multi-step diffusion models.
>
> **Q6. The connection to distribution matching distillation (Section 4.3) is valuable, but is the theoretical contribution sufficiently novel beyond applying existing NCE frameworks to this setting?**
>
> **A6.** Thanks for your comments! We think that the authors may have some misunderstanding of our method and analysis. We propose a joint energy-based model to capture the relationships among fake score models, teacher models, and one-step generators, and we train it using NCE to simplify the training pipeline—eliminating the need for iterative optimization and additional models. Furthermore, we provide a novel analysis that bridges our energy-based formulation through the perspective of distribution matching distillation. Specifically, we derive a loss from the distribution matching perspective that shares a similar form with our proposed energy-based method, offering deeper insight into why our approach is effective for distilling one-step generators. **This analysis is particularly important and is specific to our energy-based formulation, as it shows that our method implicitly optimizes a distribution matching objective—an approach that has been widely validated in prior work but typically involves a much more complex training process.**
>
>
> [1] Inductive Moment Matching.
>
> [2] Diffusion Model Alignment Using Direct Preference Optimization.
>
> [3] One Step Diffusion via Shortcut Models
>
> [4] Improved Distribution Matching Distillation for Fast Image Synthesis
>
> [5] One-step Diffusion with Distribution Matching Distillation
>
> [6] Score identity Distillation: Exponentially Fast Distillation of Pretrained Diffusion Models for One-Step Generation

---

> > ### Comment · Reviewer_deet · 2025-08-07
> >
> > Thanks for the detailed rebuttal, which has addressed some of my concerns. I recognize the contributions of this paper, but I still feel that the restricted evaluation somewhat limits the paper's quality, particularly in its lack of comprehensive evaluation on larger image resolutions and a comprehensive comparison with non-distillation approaches. Given these considerations, I will raise my score, but still with some reservations.

---

> > > ### Author Response · Authors · 2025-08-07
> > >
> > > Thank you for your additional comments and for considering a score revision! Due to limited computational resources and the time constraints of the rebuttal period, we conducted a high-resolution image experiment by fine-tuning DMD2, which demonstrates the potential of our model in more complex settings. While we leave experiments on additional datasets for future work, we think the current results are sufficient to showcase the potential of our proposed method.
> > >
> > > For non-distillation methods, we have already included a comparison on CIFAR-10 with the state-of-the-art method [1] in the rebuttal. Additionally, we now provide a comparison between the representative method cited in [1] that reports results on ImageNet using the same model size:
> > >
> > > |    ImageNet   | FID |
> > > |--------------|---------------|
> > > | ECT         |    4.05  |
> > > | ICT         |    4.02  |
> > > |      CED-DRE    |      **3.88**         |
> > >
> > > This results also verify the effectiveness of our method on ImageNet. Additionally, distillation methods show great potential in complex tasks such as text-to-image generation, where they significantly outperform non-distillation approaches as shown in [2]. This may be because the instability of non-distillation methods often requires complex techniques like time-step scheduling or converting discrete time to continuous, making them harder to train effectively. Moreover, most methods trained from scratch (non-distillation methods) tend to underperform compared to distillation-based approaches. Therefore, we think that **comparing our method with other distillation methods under the same settings provides a fair and effective way to validate our method's performance.**
> > >
> > > We would like to clarify that **our primary contribution is not to propose a state-of-the-art method for distilling multi-step diffusion models into one-step generators without real data.** Rather, our goal is to introduce a new perspective that challenges the existing complex training frameworks with extra models and iterative optimization. Our method can show great efficiency compared with distillation methods as mentioned in the rebuttal of reviwer e2GZ, **CED achieves an FID of 3.85 on CIFAR-10 using only 20 A100 GPU hours, whereas SiD requires 100 A100 GPU hours to reach an FID of 4.0.** Non-distillation methods may require a similar amount of computational resources as SiD to converge, or even more GPU hours in some cases. **We believe our method is particularly valuable in this domain, as current state-of-the-art approaches require storing, loading, and fine-tuning three separate models for complex tasks like text-to-image, making the process cumbersome and resource-intensive.** In contrast, our lightweight approach offers a more efficient and practical alternative, highlighting its importance for this field. As mentioned in our rebuttal to Reviewer x1Sm, we also explored combining our method with other techniques such as GANs, which leverage real data. **Through a simple implementation and experiment, we achieved an impressive FID of 1.85 on CIFAR-10, demonstrating the strong potential of our approach for further expansion.** We believe our method represents an important starting point for a new distillation-based technique, offering a new perspective that can drive future development and be effectively integrated with advanced methods to further enhance performance.
> > >
> > > We hope these responses help clarify our evaluation, highlight the method’s potential on complex tasks, and demonstrate its compatibility with advanced techniques. Hope that they effectively address your concerns!
> > >
> > > [1] Inductive Moment Matching.
> > >
> > > [2] Improved Distribution Matching Distillation for Fast Image Synthesis.

---

> ### Author Response · Authors · 2025-08-04
> **Dear NeurIPS Reviewer deet: we understand that you maybe busy, so we would greatly appreciate it if you could check out our rebuttal.**
>
> Dear NeurIPS Reviewer deet:
>
> We sincerely appreciate the time and thoughtful feedback you’ve provided during the review of our paper.
>
> We just want to reiterate that there are very clear-cut answers to every question and misunderstandings that were raised, and our rebuttal has carefully addressed each point-by-point.
>
> **As the discussion period nears its conclusion, we would like to ask if there are any remaining questions or points that require further clarification. If our responses have addressed your concerns, we would be grateful if you would consider updating your score accordingly.** Thank you again for your time and attention. We truly value your input.
>
> The authors of "Simple Distillation for One-Step Diffusion Models"

---

### Comment · Area_Chair_KvHD · 2025-08-05
**Reminder: Discussion and Final Justification**

Dear Reviewers,

As we approach the end of the author–reviewer discussion phase (Aug 6, 11:59pm AoE), I kindly remind you to read the author rebuttal carefully, especially any parts that address your specific comments.

Please consider whether the response resolves your concerns, and if not, feel free to engage in further discussion with the authors while the window is still open.

Your timely participation is important to ensure a fair and constructive review process. If you feel your concerns have been sufficiently addressed, you may also submit your Final Justification and update your rating early. Thank you for your contributions.

Best,

AC

---

### Note · Authors · 2025-08-12

Dear AC and reviewers:

Thank you once again for your effort in reviewing our paper—we truly appreciate your support. During the rebuttal period, **it appears that most reviewers have acknowledged our contributions and decided to raise their scores.** We would like to begin with a brief summary of our rebuttal and highlight several several important points:
- We have provided detailed explanations of our contributions to Reviewer deet, which have addressed their initial concerns. To address additional questions from Reviewer deet, **we present results comparing our approach with non-distillation methods on CIFAR-10 and ImageNet, along with high-resolution experiments that demonstrate strong potential for complex tasks.** Given that our method is greatly different from prior approaches in this domain—avoiding iterative optimization and additional components—we believe it offers valuable insights for simplifying the training pipeline, especially since models trained with previous methods are difficult to fine-tune.
- The concerns from Reviewer e2GZ mainly relate to clarification, convergence speed, and implementation details. We believe we have already addressed these points in our rebuttal.
- Our rebuttal has addressed the concerns raised by Reviewer SDf9, and we will clarify any potentially confusing words in the revised version. We believe this issue can be easily resolved.
- Reviewer x1Sm has acknowledged the value of our paper, and we addressed their concerns in the rebuttal. In particular, they noted, “A quick glance at the provided code, I see it is quite simple to implement, so this is a plus.” This aligns closely with our core contribution—offering a different perspective on simplifying the training pipeline.

In summary, our method introduces a new algorithm that avoids iterative optimization and extra components, overcoming the slower convergence, complex hyperparameters, and fine-tuning challenges of previous approaches. Therefore, we believe our simple training pipeline is an important contribution to the domain, enabling the distillation of multi-step diffusion models into a single step. We also explored advanced techniques, such as combining our approach with GANs, as demonstrated in the rebuttal, further showcasing the strong potential of our method for future work.

In the end, we are deeply grateful again to all the reviewers and the AC for engaging in the discussion and helping us improve our paper.

---

### Decision · Program_Chairs · 2025-09-17

**Decision:**

Accept (poster)

**Comment:**

This paper introduces Contrastive Energy Distillation (CED), a simple and efficient method to convert slow, multi-step diffusion models into fast, one-step image generators. The key idea is to frame the distillation as a contrastive learning task, where a student model learns to distinguish "positive" samples from a teacher model against "negative" samples from a previous version of itself. By leveraging a joint energy-based model and a contrastive loss inspired by Noise Contrastive Estimation, CED avoids auxiliary models and iterative training. It achieves competitive performance on CIFAR-10 and ImageNet.


This work has several clear strengths:
- The core idea of using a contrastive Energy objective to traina one-step generator is novel and empirically effective.

- CED eliminates the need for auxiliary score models and iterative updates, reducing memory usage and simplifying implementation.

- This work provides theoretical justification, including a compelling theoretical connection between the proposed CED objective and KL divergence.


However, the scalability concern is one major remaining concern. The scalability to more complex data generation, such as larger resolution and even videos, is unclear. Moreover, it is also difficult to understand CED with model size, given the existing results. I partially agree with Reviewer deet’s final justifcation that the restricted evaluation somewhat limits the paper's quality, particularly in its lack of comprehensive evaluation on larger image resolutions and a comprehensive comparison with non-distillation approaches.

While some concerns remain, I believe this work is novel and interesting enough for being accepted to NeurIPS. It will promising to further explore its scalability to larger models and more complex data.

In summary, I recommend Accept (Poster).